# BENCHMARKS FOR REINFORCEMENT LEARNING WITH BIASED OFFLINE DATA AND IMPERFECT SIMULATORS

## ABSTRACT

In many reinforcement learning (RL) applications one cannot easily let the agent act in the world; this is true for autonomous vehicles, healthcare applications, and even some recommender systems, to name a few examples. Offline RL provides a way to train agents without real-world exploration, but is often faced with biases due to data distribution shifts, limited coverage, and incomplete representation of the environment. To address these issues, practical applications have tried to combine simulators with grounded offline data, using so-called *hybrid methods*. However, constructing a reliable simulator is in itself often challenging due to intricate system complexities as well as missing or incomplete information. In this work, we outline four principal challenges for combining offline data with imperfect simulators in RL: simulator modeling error, partial observability, state and action discrepancies, and hidden confounding. To help drive the RL community to pursue these problems, we construct "Benchmarks for Mechanistic Offline Reinforcement Learning" (B4MRL), which provide dataset-simulator benchmarks for the aforementioned challenges. Our results show that current algorithms fail to synergize these sources, often performing worse than using one source alone, especially when faced with hidden confounding.

## 1 INTRODUCTION

Reinforcement learning (RL) is a learning paradigm in which an agent explores an environment in order to maximize a reward Sutton & Barto (2018). However, in many applications exploration can be costly, risky, slow, or impossible due to legal or ethical constraints. These challenges are evident in fields such as healthcare, autonomous driving, and recommender systems.

To overcome these obstacles, two principal methodologies have emerged: using offline data, and incorporating simulators of real-world dynamics. Both approaches have distinct advantages and drawbacks. While offline data is sampled from real-world dynamics and often represents expert policies and preferences, it is limited by exploration and finite data Levine et al. (2020); Fu et al. (2020); Jin et al. (2021). Furthermore, offline data often suffers from confounding bias, which occurs when the agent whose actions are reflected in the offline dataset acted based on information not fully present in the available data: For example, a human driver acting based on eye-contact with another driver, or a clinician acting based on an unrecorded visual inspection of the patient. Confounding can severely mislead the learning agent Zhang & Bareinboim (2016); Gottesman et al. (2019); De Haan et al. (2019); Wang et al. (2021), as we demonstrate in our paper. We refer to these sources of error as *offline2real*.

In contrast to learning from offline data, simulators allow nearly unlimited exploration, and have been the bedrock of several recent triumphs of RL (Mnih et al., 2013; Vinyals et al., 2019; Wang et al., 2023). However, utilizing simulators brings its own set of challenges, most notably – modeling error. This error often arises due to the complexity of real-world dynamics and the inevitability of missing or incomplete information. Although simplified simulators are widely used, any discrepancies between their dynamics and real-world dynamics can lead to unreliable predictions. These so-called *sim2real* gaps may range from misspecifications in the transition and action models to biases in the observation functions (Abbeel et al., 2006; Serban et al., 2020; Kaspar et al., 2020; Ramakrishnan et al., 2020; Arndt et al., 2020).

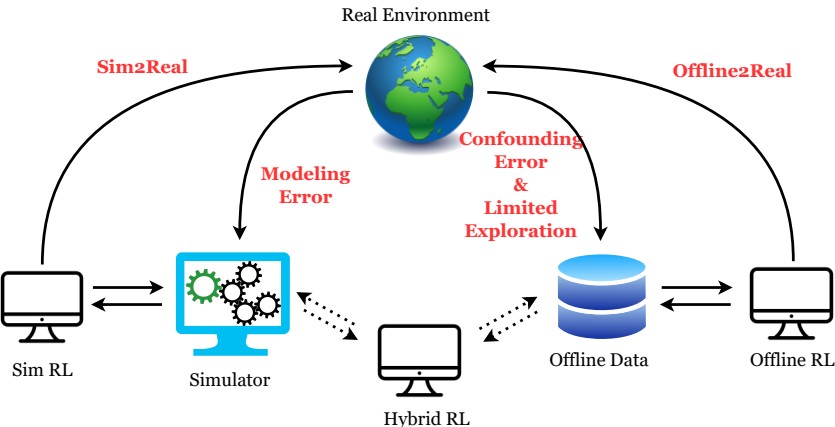

Figure 1: An illustration of the discrepancies and biases arising when training RL agents. *Modeling error* refers to the discrepancy between the real world dynamics and the simulator, e.g. transition error. *Confounding error* refers to bias due to the dataset not including factors affecting the behavioral policy. Other challenges include limited exploration, partial observability and state and action discrepancies, as detailed in Section 2.

In recent years, there has been a growing recognition of the complementary strengths and limitations of offline data and simulation-based approaches in RL (Nair et al., 2020; Song et al., 2022; Niu et al., 2022; 2023). Recent work has merged these two approaches to leverage their respective advantages and mitigate their drawbacks; namely, offline data, which provides real-world expertise and preferences, with simulators which offer extensive exploration capabilities. These hybrid methods hold promise for addressing the challenges posed by costly or limited exploration in various domains.

However, evaluating hybrid RL methods requires standardized benchmarks that systematically capture the distinct biases inherent in both offline data and simulators. A recent effort in this direction is ODRL (Lyu et al., 2024b), which introduced a benchmark suite incorporating dataset variants with different simulator misspecifications to assess offline, online, and hybrid RL performance. While ODRL represents a significant step forward, it focuses only on simulator discrepancies and does not comprehensively cover the full spectrum of challenges encountered in hybrid RL.

In this work, we present four key challenges for merging simulation and offline data in RL: modeling error, partial observability, state and action discrepancies, and confounding error. We propose a set of benchmarks to systematically explore hybrid RL approaches, termed "Benchmarks for Mechanistic Offline Reinforcement Learning" (B4MRL). Each benchmark reflects differences between simulators and offline data. Table 2 compares B4MRL to existing benchmarks, highlighting their limitations and showing that B4MRL uniquely addresses all four challenges (see Appendix A for details). We demonstrate how contemporary hybrid and offline RL approaches can fail when confronted with these challenges, suggesting the necessity of our benchmarks for future research.

## 2 CHALLENGES OF COMBINING OFFLINE DATA WITH SIMULATORS

In this section, we outline the four key challenges. We present a systematic taxonomy of these discrepancies in Table 1, organizing them by their nature and primary source.

### 2.1 MODELING ERROR (SIM2REAL)

Simulators, as computational representations of real-world systems, inherently contain modeling errors. These errors arise from simplifications and assumptions made during the simulator's design and construction to render the simulation manageable and computationally tractable, a process which often introduces systemic differences or biases between the simulator's dynamics and the real-world system. For example, a weather simulation may be biased due to an imperfect understanding of atmospheric dynamics, and a diabetes simulation might not accurately simulate the complexities of the

Table 1: A Summary of the Four Core Challenges in B4MRL. This taxonomy categorizes the discrepancies we study.

| Challenge | Definition | Source | Concrete Example |
|---|---|---|---|
| **1. Modeling Error**[*] | The simulator's dynamics do not perfectly match the real world. | Sim2Real | A diabetes simulation fails to accurately model the human body's complex reaction to exercise. |
| **2. State Discrepancy** | The state representation differs between sources or is incomplete (partial observability). | Sim2Real Offline2Real | An autonomous driving simulator omits subtle pedestrian gestures that signal intent to cross the road. |
| **3. Action Discrepancy** | The action representation differs between sources. | Sim2Real | A simulator has a discrete `lane_change` action, while real data has continuous steering angles. |
| **4. Hidden Confounding** | Unobserved factors in offline data influenced both the actions and outcomes. | Offline2Real | A doctor's treatment choice is based on a visual cue not recorded in the patient's electronic health record. |

[*]Modeling Error can manifest as flawed dynamics (our experimental focus) or a flawed reward function.

body's reaction to exercise. Consequently, these biases can influence the decisions and actions taken by a reinforcement learning agent trained on such simulators, leading to suboptimal performance when transferred to the real world.

## 2.2 Partial Observability and State Discrepancy (Sim2Real, Offline2Real)

Simulators are often designed to abstract and simplify real-world complexities, selectively modeling aspects of a problem that are most relevant to the intended application. This selective modeling can create blind spots as parts of the real-world observation space are omitted or oversimplified. For example, consider an autonomous driving simulator. It might accurately model the dynamics of vehicles and pedestrian movement. However, to keep the simulator manageable and tractable, it may exclude details such as subtleties of human behavior, including facial expressions or gestures that could signal an intent to cross the road. Despite these omissions, the simulator remains a valuable tool for training autonomous driving systems. However, its partial state description can lead to biases in the learned policy, which might be suboptimal or even erroneous in the real world.

Similarly, in Offline2Real scenarios, data collected from real-world environments might suffer from partial observability due to constraints in the data collection process or limitations in sensor technology. For instance, in healthcare settings, electronic health records might not capture information about a patient's mental state or genetics, which can significantly influence health outcomes. Partial observability in offline data may or may not lead to confounding bias, as we discuss in Section 2.4.

## 2.3 Action Discrepancy (Sim2Real)

One of the substantial challenges in merging simulation and offline data lies in inconsistencies between action definitions in simulation environments and offline data. Every action taken by an agent in the real environment can be nuanced and multifaceted. Simulators, on the other hand, have to abstract these complexities into a more manageable and computationally feasible representation. As a result, there can be a disconnect in how actions are represented in these two different systems. For example, in an autonomous driving system, the action might be discrete and only choose between moving a lane to the left or staying in the current lane. However, in real-world data, the actions might also include more specific information like the exact amount of torque change, and the steering angle.

## 2.4 Confounding Bias (Offline2Real)

The presence of unobserved (hidden) confounding variables poses a significant challenge when using observational data for decision-making. Hidden confounding occurs when in the process that generated the offline data, unobserved factors influenced both the outcome and the decisions made by the agent. This can lead to unbounded bias, a result which is well known from the causal inference

Table 2: Comparison of benchmarks in terms of suitability for online/offline algorithms and their ability to model various real-world challenges. There are only two benchmarks that deal with Hybrid-RL scenarios, and this work is the only one that deals with all four challenges.

| Benchmark | Online/ Offline | Simulator Modeling Error | State Discrepancies | Action Discrepancies | Hidden Confounding |
|---|---|---|---|---|---|
| **B4MRL (This Work)** | **Hybrid** | ✓ | ✓ | ✓ | ✓ |
| D4RL Fu et al. (2020) | Offline | ✗ | ✗ | ✗ | ✗ |
| VD4RL Lu et al. (2022) | Offline | ✗ | ✓ | ✗ | ✗ |
| ODRL Lyu et al. (2024b) | Hybrid | ✓ | ✗ | ✗ | ✗ |
| CARL Benjamins et al. (2021) | Online | ✓ | ✓ | ✗ | ✗ |
| Gym-extensions Henderson et al. (2017) | Online | ✓ | ✗ | ✗ | ✗ |
| RLBench James et al. (2019) | Online | ✗ | ✗ | ✗ | ✗ |
| DMC Suite Tassa et al. (2018) | Online | ✗ | ✗ | ✗ | ✗ |
| Continual World Wołczyk et al. (2021) | Online | ✗ | ✗ | ✗ | ✗ |
| Meta-World Yu et al. (2020a) | Online | ✗ | ✗ | ✗ | ✗ |

literature Pearl (2009); Zhang & Bareinboim (2016); Tennenholtz et al. (2020; 2022); Uehara et al. (2022); Hong et al. (2023). This issue becomes particularly pertinent in sequential decision-making scenarios and can substantially impact the performance of learned policies. Hidden confounding is prevalent in diverse real-world applications, including autonomous driving, where unobserved factors like road conditions affect the behavior of the human driver, and healthcare, where for example unrecorded patient information or patient preferences may influence the decisions made by physicians as well as patient outcome. In Figure 2 we show a POMDP with hidden confounding.

Effectively addressing hidden confounding in offline RL is paramount to ensure the reliability and effectiveness of learned policies. Research has attempted to develop methodologies to account for confounding bias, including: the identification of hidden confounders using interventions or extra data sources (Angrist et al., 1996; Jaber et al., 2018; Lee & Bareinboim, 2021; von Kügelgen et al., 2023; Kallus et al., 2018; Zhang & Bareinboim, 2019; Tennenholtz et al., 2021; Lee et al., 2020), and the quantification and integration of uncertainty arising from confounding into the learning process (Pace et al., 2023). We believe there is a crucial need for benchmarks and datasets designed to address this issue, enabling researchers to compare and evaluate different methods for handling confounding bias in offline RL. We emphasize that hidden confounding and partial observability are distinct concepts. While they intersect in some cases, it is crucial to recognize their differences to effectively address their challenges, as we demonstrate in the following example.

To demonstrate the impact of hidden confounding bias in offline RL, consider the following single-state decision problem with two actions $\{a_0, a_1\}$. We let $z \in \{0, 1\}$ such that $P(z = 0) = \frac{1}{3}$, and $P(z = 1) = \frac{2}{3}$. Additionally, let the reward $r \in \{0, 1\}$, such that $P(r = 1|z = 1, a = a_1) = \frac{1}{2}$, $P(r = 1|z = 1, a = a_0) = \frac{1}{3}$, $P(r = 1|z = 0, a = a_1) = \frac{1}{4}$ and $P(r = 1|z = 0, a = a_0) = \frac{1}{6}$. Note that action $a_1$ dominates, and with or without access to $z$ at decision time the optimal action is given by $a^* = a_1 = \arg\max_a \mathbb{E}_{z \sim P(z)} P(r = 1|z, a)$.

Next, let $\pi_b(a|z)$ be some behavioral policy (with access to $z$), which deterministically selects action $a_1$ when $z = 0$ and selects action $a_0$ when $z = 1$. We ask, can data generated by $\pi_b$ be used to learn a good policy if $z$ is not provided in the data? That is, can we learn a policy which maximizes $\mathbb{E}_{z \sim P(z)}[P(r = 1|a, z)]$? Unfortunately, $z$ acts as a hidden confounder, which significantly biases our results, even in the limit of infinite data. Indeed, our data is sampled from $P^{\pi_b}(r, a) = \mathbb{E}_{z \sim P(z)}[P(r|a, z)\pi_b(a|z)]$, and thus $P^{\pi_b}(r = 1|a = a_0) = \frac{\mathbb{E}_{z \sim P(z)}[P(r=1|a_0,z)\pi_b(a_0|z)]}{\mathbb{E}_{z \sim P(z)}[\pi_b(a_0|z)]} = P(r = 1|a_0, z = 1) = \frac{1}{3}$. Similarly, $P^{\pi_b}(r = 1|a = a_1) = P(r = 1|a_1, z = 0) = \frac{1}{4}$. Therefore, even in the limit of infinite data, the standard empirical estimator $\hat{\pi} \in \arg\max_a P^{\pi_b}(r = 1|a)$ would yield a suboptimal result of select-

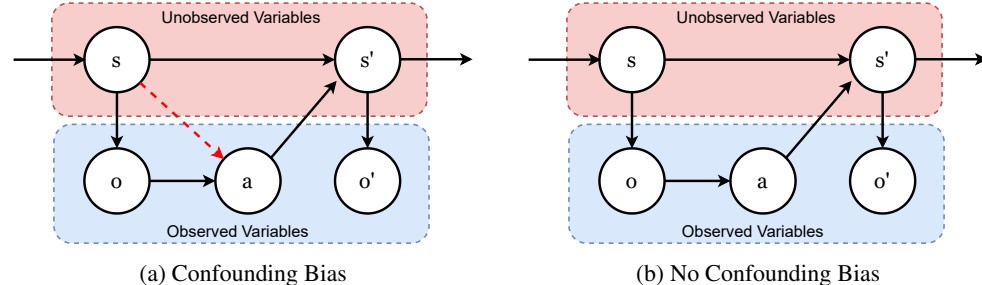

(a) Confounding Bias         (b) No Confounding Bias

Figure 2: Two causal graphs of POMDPs. While in both cases the state $s$ is not observed, only in figure (a) $s$ acts as confounder, as actions in the data were taken w.r.t. the unobserved $s$.

ing action $a_0$. This error is due to the dependence of both $a$ and $r$ on the hidden confounder $z$, and not only the fact that it is unobserved. Moreover, this bias cannot be mitigated with increasing the number of samples, unlike the statistical uncertainty induced by finite data.

In the next section, we shift our focus to developing benchmarks that serve as a rigorous testing ground for RL algorithms. These benchmarks were designed to illuminate the aforementioned challenges, helping researchers devise strategies to mitigate them, thereby promoting the advancement of robust, reliable, and high-performing RL systems that effectively utilize both offline data and simulations.

## 3   BENCHMARKS FOR MECHANISTIC OFFLINE REINFORCEMENT LEARNING (B4MRL)

In this section, we outline the "Benchmarks for Mechanistic Offline Reinforcement Learning" (B4MRL), designed for evaluation of RL methods using both offline data and simulators, which we refer to as hybrid algorithms. The proposed datasets and simulators encompass a range of discrepancies between the true dynamics, the simulator, and the observed data.

Given the four principal challenges delineated in Section 2 – namely, modeling error, partial observability, discrepancies in states and actions, and confounding bias – we created benchmarks based on the MuJoCo robotic environment (Todorov et al., 2012), and the Highway environment (Leurent, 2018). The MuJoCo tasks are popular benchmarks used to compare both offline RL and online RL algorithms, including multiple environments: HalfCheetah, Hopper, Humanoid, Walker2D. These environments provide the agent observations of variables describing the controlled robot such as the angle and angular-velocity of the robot joints, and the position and velocity of the different robotic parts (e.g., an observation in HalfCheetah consists of 17 variables). The acting agent can perform actions at a given time by applying different torques to each joint (e.g. in HalfCheetah there are 6 joints, hence an action consists of 6 continuous variables). The reward function differs between the different tasks, and relies mainly on the speed and balance of the robot. The Highway environment simulates the behavior of a vehicle aiming to maintain a high speed while avoiding collisions. The observations include the current position and velocity of the controlled vehicle and the other vehicles on the road, and lets the agent control the throttle and steering angle of the controlled vehicle.

In recent years several MuJoCO-based offline-RL benchmarks and datasets emerged, offering different characteristics and challenges. The most common one, and the one we build upon in this paper, is the Datasets for Deep Data-Driven Reinforcement Learning benchmark, or D4RL (Fu et al., 2020). These datasets are categorized by scores achieved by an underlying data-generating-agent, ranging from completely random agents, to "medium" level agents, through expert agents, and further provide datasets with heterogeneous policy mixtures (e.g., medium-expert). We note that by construction, these datasets do not suffer from hidden confounding. Our work builds upon and expands these datasets by implementing imperfect simulators and the other challenges outlined in Section 2. While the aim of this paper is to provide benchmarks for hybrid-RL algorithms, we stress that the benchmarks we provide in some of the challenges could also be used to test offline-RL and online-RL algorithms. We constructed these benchmarks such that researchers can easily create new benchmarks for evaluating the various challenges. Exhaustive details in Section C.

**Challenge 1: Modeling Error.** We induce modeling error by introducing changes in simulator dynamics which directly influence the transition function over time. Small errors in transition dynamics could aggregate to produce completely wrong state predictions over long horizons. Specifically, in this benchmark we propose changing one of the environment parameters that affects the simulator's dynamics. For example, in the HalfCheetah and Walker environments, we propose two benchmarks: changing the gravitation parameter to $g_{\text{sim}} = 19.6$ instead of $9.81$, and changing the friction parameter by multiplying it by a factor of $0.3$.

**Challenge 2: Partial Observability and State Discrepancy.** We implement this challenge with two primary mechanisms: (1) Structural Discrepancy, where key variables are hidden from the agent's observation. We provide two such benchmarks ($h_{\text{low}}$ and $h_{\text{high}}$), chosen after an ablation study (see Figure 4), which remove a specific variable from the observations. (2) Observational Noise, a simpler, non-structural case where we add Gaussian noise ($\sigma_{\text{low}}$ and $\sigma_{\text{high}}$) to the full state. Combining these options with the four D4RL datasets yields 16 benchmarks per environment.

In addition to benchmarks with partial observability in the simulator, we add a complementary benchmark with partial observability in the dataset. This is achieved by creating a new dataset with a data generating agent that trains and collects data on partially observed environment states. To form this benchmark we created two new datasets, each missing a different variable. Specifically, we removed the same variables $h_{\text{low}}$ and $h_{\text{high}}$, as described above. For the hybrid-RL algorithms we combine the new datasets with a simulator suffering from transition error, resulting in a total of two benchmarks. Importantly, while these datasets suffer from partial observability, they do not suffer from hidden confounding, as the data-generating agent decides on its next action based on the same observation that is registered in the data; see Figure 2b.

**Challenge 3: Action Discrepancy.** The third challenge centers around the issue of discrepancies between actions. To allow evaluation of the impact of action errors we altered how actions taken by the agent in the simulator state dictate the transition to the next state. To that end, we introduce Gaussian noise to the action before it is passed to the simulator's transition function, while the dataset's actions remain without noise. This creates a discrepancy between the simulator and the data in the effect of actions on the state. We benchmark the models on two noise levels: noise with low variance $\sigma_{\text{low}}$, and noise with high variance $\sigma_{\text{high}}$. As before, the choice of values was done based on the results of the SAC algorithm on the noisy simulator. The benchmark includes the combination of the 4 D4RL datasets and a simulator with action discrepancy (low noise, high noise) resulting in 8 different datasets.

**Challenge 4: Confounding Bias.** For this challenge, we assume we do not have complete access to the state that the data generating agent utilized when determining its actions. This is a special and important case of partial observability which occurs in offline data and can induce bias due to the behavior policy's dependence on the unobserved factors, see Section 2.4.

For this benchmark we build on the D4RL datasets as follows: We either add Gaussian noise to the observations in the data, or we omit a dimension recorded in the dataset observations. This is fundamentally different from the partial observability in Challenge 2. Here, the unobserved information was used by the data-generating agent to make decisions, creating a spurious correlation between the (incomplete) observations and the actions, as shown in Figure 2. Thus, the data generating agent decided on action $a_i$ based on the full system state $s_i$, but we have access only to a noisy or projected observation $o_i$; see Figure 2a. This creates a dataset with hidden confounding, where we do not have full information on why a specific action was chosen.

As established in causal inference (Pearl, 2009; Zhang & Bareinboim, 2016; Tennenholtz et al., 2020; 2022; Uehara et al., 2022; Hong et al., 2023), this confounding can incur arbitrary bias, which we now demonstrate experimentally. We used the same settings as the observation-error benchmark: low and high Gaussian noise on the observations in the data ($\sigma_{\text{low}}$ and $\sigma_{\text{high}}$), and missing dimensions ($h_{\text{low}}$ and $h_{\text{high}}$) from the observations in the data, resulting in 16 benchmarks.

Finally, we provide a benchmark for confounding by creating a new dataset where the data-generating-agent acts based on a history of three observations, instead of the last one. Hiding the fact that the dataset actions were history-aware can induce hidden confounding. For this benchmark we create history-aware dataset with hidden variables ($h_{\text{low}}$ and $h_{\text{high}}$), and use a simulator with transition error.

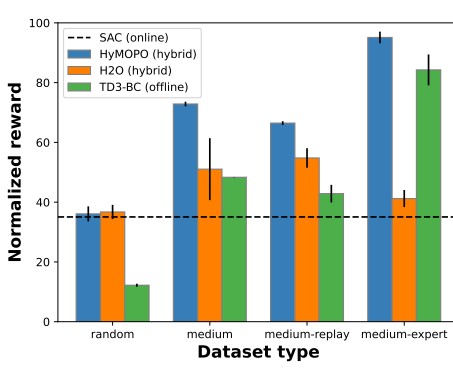
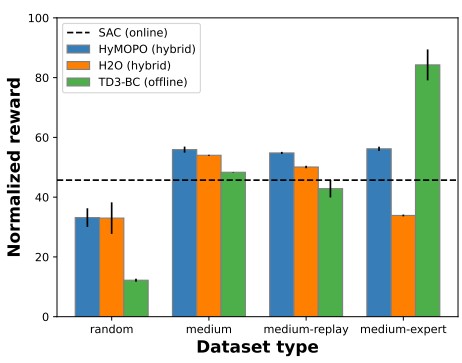

(a) Simulator with transition error

(b) Partially observed simulator

Figure 3: Results on HalfCheetah environment for modeling error and partial observability. In both figures, the algorithms have access to the standard D4RL datasets, but use different types of imperfect simulators. For modeling error **(a)** we introduced an error in the transition function by setting the gravitational parameter to $g = 19.6$ instead of $9.81$, and for partial observations **(b)** we added Gaussian noise ($\sigma = 0.05$) to the full state.

As explained above, the proposed set of benchmarks can be used to evaluate offline-RL algorithms as well as hybrid-RL algorithms, as it poses the problem of confounded datasets that do not have a standardized benchmark. For hybrid-RL algorithms, we use an imperfect simulator with transition error (as described in challenge 1), along with the dataset benchmarks described in this challenge.

To facilitate standardized comparison and prevent configuration ambiguity in future work, we formally define the B4MRL Core Evaluation Protocol in Section B. We recommend that practitioners utilize these specific high-intensity settings to rigorously stress-test algorithmic robustness.

## 4 EXPERIMENTS

In this section we present empirical evaluations following the procedures described in Section 3 above. We used online, offline, and hybrid RL methods to showcase challenges and limitations in current RL approaches for hybrid tasks. Our array of methods represents a cross-section of state-of-the-art RL approaches in both model-based and model-free paradigms, providing a broad look at how diverse techniques perform in the face of our hybrid RL benchmarks. For demonstration purposes, the experiments in the main text focus on the HalfCheetah environment; however, we also conducted experiments on additional environments, yielding similar results, which can be found in section G.

### 4.1 BASELINES

To evaluate the effectiveness of our proposed benchmarks, we selected a set of online, offline, and hybrid RL algorithms. These algorithms have been used extensively in numerous RL papers, and shown to successfully achieve high and reliable rewards. For online RL we used **TD3** (Fujimoto et al., 2018) and **SAC** (Haarnoja et al., 2018). For offline-RL algorithms, we used the model-based **MOPO** (Yu et al., 2020b), as well as the model-free approaches **TD3-BC** (Fujimoto & Gu, 2021) and **IQL**(Kostrikov et al., 2021). Finally, to test hybrid-RL algorithms, we used three algorithms that can jointly use both a simulator and offline data: the **H2O** (Niu et al., 2022) algorithm, a behavioral-cloning variant of the PAR (Policy Adaptation by Representation mismatch) algorithm named **PAR-BC** (Lyu et al., 2024a), and a variation of MOPO we term **HyMOPO**, for Hybrid-MOPO (model based offline policy optimization). H2O adaptively adjusts Q-values on simulated data according to the dynamics gap evaluated against real data. PAR penalizes the source domain (the simulator in our case) data by measuring the representation mismatch between two domains (the simulator, and the data domains), and its BC variant introduces an additional BC term. HyMOPO is similar to MOPO but includes several key modifications: Standard MOPO trains a dynamics model on the offline dataset $\mathcal{D}$ to predict the next observation $o'$ and reward $r$ given the current

observation $o$ and action $a$. HyMOPO can also access a simulator, so it first queries the simulator for its prediction of the next observation $o'$sim for each observation-action tuple in $\mathcal{D}$. Next, HyMOPO learns a dynamics model $f$ such that $o' = o'_{\text{sim}}(a, o) + f(a, o)$. Thus, HyMOPO's goal is to learn an additive function that corrects the gap between the simulator's prediction and the dataset's next observation. The remaining steps are the same as in MOPO. For full details, see Section D.

This selection of algorithms is designed to explore the critical distinction between data quality and data synergy. Significant prior work in Robust Offline RL focuses on the data quality problem: how to learn from a single, static, and corrupted dataset. Our benchmark, in contrast, evaluates the data synergy problem: how can an agent best arbitrate between two imperfect sources, a flawed simulator and flawed offline data? To this end, our baselines include IQL, an algorithm known to be highly robust to data corruption, to demonstrate that even strong offline-only methods are insufficient by themselves to solve the unique, synergistic challenges of the hybrid setting.

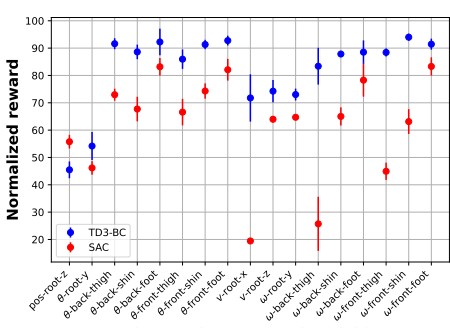

Figure 4: Results of offline (TD3-BC) and online (SAC) algorithms on the HalfCheetah environment with a single missing variable. TD3-BC runs on the medium-expert dataset. For each label on the x-axis, SAC trained on partially observed simulator that lacks that variable, and TD3-BC trained on a dataset that did not have any information about that variable, despite it being used by the agent which generated the dataset.

### 4.2 RESULTS

We benchmarked the challenges detailed above on the MuJoCo-HalfCheetah environment, and present here the main results. Full details and more results can be found in Section G. We report results as mean and standard deviation of the normalized rewards (scaling raw rewards to a scale of 0 (random) to 100 (expert) as in D4RL) across three random seeds.

In Figure 3 we show how different RL approaches perform on the modeling error challenge. Both hybrid algorithms, HyMOPO and H2O, demonstrate an interesting phenomenon. First, as expected, on the medium and medium-replay datasets both methods score better than SAC (online-RL), which uses only the simulator, and TD3-BC (offline-RL) which uses only the datasets. However, when using the simulator with observation error and the random dataset, we observed both hybrid-RL algorithms scored *worse* than only using SAC on the simulator – unexpectedly, using the offline dataset negatively impacted the hybrid approaches. We observed the same phenomenon in other cases as well. For example, in the medium-expert dataset with a partially observable simulator, HyMOPO scored less than TD3-BC trained on the data alone, and H2O scored even worse, being inferior to both SAC on the simulator alone and TD3-BC on the dataset alone.

In Figure 4, we demonstrate the effect of hidden confounders by comparing an online algorithm (SAC) on a partially observable simulator and an offline algorithm (TD3-BC) on the medium-expert dataset with hidden confounders. In the online case, the algorithm has access to the full state except for a single dimension, and in the offline case, we remove the exact same variable from the dataset, even though it was used by the agent generating the data. Note that algorithms that do not use offline data cannot suffer from hidden confounding, though they may suffer from partial observability. We trained both algorithms with each possible variable removed (one at a time) and compared the results. While one might expect the importance of a variable $v$ for performance in the online algorithm to be similar to its importance in offline learning, we show that some variables are more important in the offline case. For example, `pos-root-z` (the $z$ coordinate of the front tip) significantly affects offline TD3-BC, while `v-root-x` (the $x$ coordinate velocity of the front tip) significantly affected online SAC. This suggests that variable `pos-root-z` induces strong hidden confounding, significantly affecting the reward as well as the choice of actions by the data-generating-agent.

To evaluate the impact of hidden confounding, our most complex challenge, we combine a flawed simulator (with gravity error) with confounded offline datasets. To provide a high-level summary of

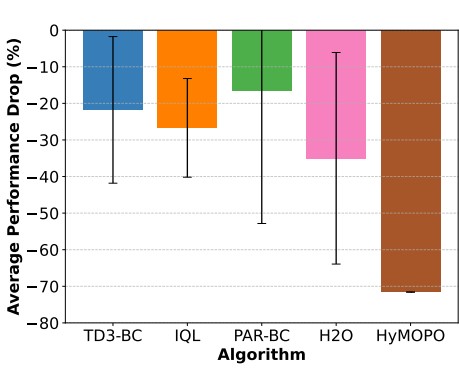
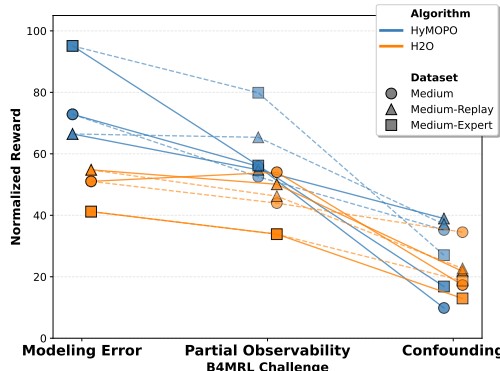

(a) Algorithm Robustness to High-Impact Hidden Confounding (avg. drop).

(b) Per-Challenge Performance Breakdown (HalfCheetah).

Figure 6: The Impact of Hidden Confounding. (a) Average performance drop across three MuJoCo environments when moving from simple modeling error to high-impact hidden confounding (Challenge 4 + 1). (b) A detailed breakdown on HalfCheetah, showing that while algorithms handle Challenge 1 (modeling error) and 2 (partial obs.) reasonably, they suffer a severe performance collapse when faced with Challenge 4 + 1 (confounded dataset and simulator with modeling error). Each line corresponds with one of the specific challenges (noise or hidden variables).

these results, Figure 6(a) plots the average performance degradation for our key algorithms on the challenging medium-expert dataset.

This aggregated view provides a quantitative summary of the key findings. It clearly visualizes the severe performance degradation that most offline and hybrid algorithms suffer, a central conclusion of our work. Furthermore, it highlights the more nuanced results, such as the surprising average robustness of TD3-BC and the complex, environment-specific interactions exhibited by hybrid methods like H2O and PAR-BC.

We further demonstrate the importance of data confounding in Figure 6(b), in which we compare two of the hybrid algorithms on 3 different challenges. Both HyMOPO and H2O achieve decent results on challenges 1 and 2, but suffer severely when encountering data confounding in challenge 4. While some degradation is expected (as the algorithms face two challenges), the severity of the collapse demonstrates the crucial effect of hidden confounding on the algorithms. Especially when both algorithms have already shown to bypass the transition error when it is present in the simulator, and that the exact same $\sigma_{\text{high}}$ and $h_{\text{high}}$ were used as in challenge 2 and in challenge 4 (missing in the simulator and in the dataset respectively).

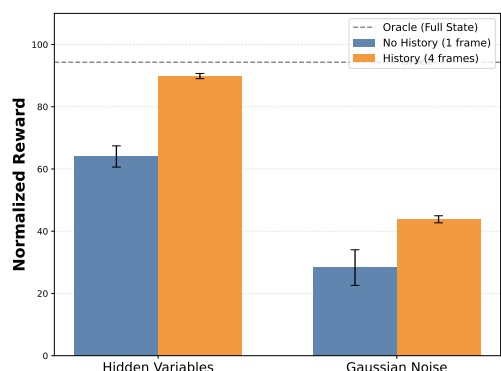

Figure 5: **Structural Analysis (IQL).** Memory ($k = 4$) recovers $\approx 95\%$ of performance in *Hidden Dimensions* (Ch. 2) but fails on *Mechanistic Noise* (Ch. 4), proving the latter is a structural barrier.

For the online simulator we used a simulator with transition error in the gravitational parameter ($g = 19.6$). Under low confounding, HyMOPO scored best across all options except the random dataset with $h_{\text{low}}$, where the simulator alone performed slightly better. Under high-confounding, both hybrid models and MOPO suffered severely. Interestingly, on the medium-expert dataset, which is twice as big as the medium dataset and has access to *optimal* trajectories, these algorithms' scores diminish, emphasizing the negative effects of hidden confounders in the data even on hybrid methods.

While hybrid algorithms are expected to perform at least as well as the best between online and offline approaches, our results reveal this can be far from reality. We further identify hidden confounding as a significant issue for the performance of offline methods.

Table 3: **Evidence of Signal Dominance.** Comparisons of Value Estimation Error ($Q_{pred}/G_{real}$). In the failure case (High Confounding), the agent overestimates returns by $> 300\%$, indicating it prioritizes the biased simulator over grounded data.

| Setting | Challenge Type | Return ($G$) | Pred. $Q$ | Ratio |
|---|---|---|---|---|
| Baseline (Healthy) | Modeling Error (Ch 1) | 11,500 | 882 | **0.077** |
| Baseline (Healthy) | Low Confounding ($\sigma_{low}$) | 9,664 | 752 | **0.078** |
| Control (SAC) | Hidden Dims (Ch 2) | 2,254 | 178 | **0.079** |
| **Failure Case** | **High Conf. ($\sigma_{high}$)** | **1,538** | **395** | **0.269** |

### 4.3 DIAGNOSIS OF FAILURE MECHANISMS

To distinguish between failures caused by limited context (resolvable via memory) and fundamental objective mismatches, we performed two diagnostic analyses.

**Is Confounding Reducible to a POMDP?** We first tested if Challenge 4 acts merely as partial observability, resolvable by history. We evaluated IQL with frame stacking (by stacking four observations together) on Challenge 4 with hidden dimensions ($h_{high}$) and with mechanistic noise ($\sigma_{high}$), on HalfCheetah Medium-Expert dataset. Figure 5 shows that history yields substantial recovery in the Hidden Dimensions setting (89.9, near the 94.3 oracle, which is the IQL score without confounding), confirming it behaves as a solvable POMDP. In contrast, frame stacking yields only marginal gains on the mechanistic noise confounding ($28.3 \rightarrow 43.8$). This implies that mechanistic noise creates an unstructured discrepancy at each timestep, acting as a structural barrier standard recurrent architectures cannot resolve.

**Mechanism of Hybrid Failure: Signal Dominance.** Next, we investigated the failure mechanism of the hybrid algorithm HyMOPO, by analyzing the Value Estimation Error (the ratio between the predicted $Q$ value and the total return). Table 3 shows that in "healthy" runs, the agent maintains a calibrated ratio of $\approx 0.08$. However, in the failure case, this ratio spikes to $0.269$ ($> 300\%$ overestimation). This supports the hypothesis of *Signal Dominance*: faced with conflicting supervision, the hybrid agent prioritizes the "clean" signal of the biased simulator over the "noisy" signal of the offline data, leading to value hallucination.

These findings validate the utility of B4MRL as a diagnostic suite: by isolating specific error sources, it enables researchers to distinguish between memory-based failures and fundamental objective misalignments, a level of diagnostic precision often impossible in unstructured real-world data.

## 5 CONCLUSIONS AND FUTURE WORK

In this paper, we provide insights into the challenges encountered when combining offline data with imperfect simulators in reinforcement learning (RL). Our newly introduced **B4MRL** benchmarks facilitate the evaluation and understanding of these complexities, highlighting four main challenges: simulator-modeling error, partial observability, state-action discrepancies, and confounding bias.

Our results reveal that current hybrid methods that combine simulators and offline datasets do not always lead to superior performance, pointing to an important future research direction. In addition, hidden confounders in the dataset can significantly affect the performance of all tested methods, including hybrid ones. In light of these results, we suggest that future work focus on developing more robust hybrid RL algorithms that can better handle modeling errors and hidden confounders, and that perform at least as well as either simulator-based methods or offline learning alone. We further discuss limitations and broader impact in Section E and Section F, respectively.

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

## A    COMPARISON TO OTHER BENCHMARKS

Prior benchmark suites for reinforcement learning typically focus on either the online or offline setting, often isolating specific challenges such as modeling error or partial observability, as summarized in Table 2. For example, D4RL Fu et al. (2020) and VD4RL Lu et al. (2022) are exclusively designed for offline RL and do not address discrepancies between simulation and real-world data. VD4RL provides pixel-based observations, which can be used in conjunction with simulators that expose ground-truth state features, allowing for limited exploration of state discrepancies—though this is not an intended feature of the benchmark.

Gym-extensions Henderson et al. (2017) was designed for multi-task transfer learning and includes a range of related tasks. It could be adapted to introduce modeling error by adding perturbations to simulator dynamics or parameters. Similarly, CARL Benjamins et al. (2021) allows the user to control contextual variables at each episode, and could be modified to simulate modeling error by introducing parameter shifts between training and evaluation. However, neither benchmark was designed to evaluate modeling error, and neither addresses offline RL or supports hybrid scenarios that combine simulators and offline data in a principled manner.

ODRL Lyu et al. (2024b) is the only prior benchmark that explicitly targets hybrid RL, but it focuses solely on modeling error and does not systematically incorporate state discrepancies, action mismatches, or hidden confounding—factors that, as we demonstrate in Section 4, have a substantial impact on performance. It continues the line of work introduced in H2O Niu et al. (2022), which proposed an algorithm for hybrid RL and included a small set of modeling error benchmarks. ODRL expands on this by providing a broader suite of modeling error scenarios, though it still does not address the other key challenges.

In contrast, B4MRL is the first benchmark suite designed to comprehensively evaluate hybrid RL methods by simultaneously addressing four critical challenges: modeling error, state discrepancy, action discrepancy, and hidden confounding. This breadth enables B4MRL to more faithfully reflect the complexities of real-world decision-making systems, where simulators and offline data could be jointly leveraged, and sets a new standard for evaluating hybrid RL algorithms.

## B    THE B4MRL CORE EVALUATION PROTOCOL

To facilitate standardized comparison and prevent configuration ambiguity in future work, we formally define the **Core Evaluation Protocol**. We recommend that practitioners reporting results on B4MRL utilize the specific configurations listed in Table 4. This protocol focuses on the *High Intensity* settings for each challenge to rigorously stress-test algorithmic robustness.

A complete B4MRL evaluation consists of reporting the Normalized Score on the HalfCheetah, Hopper, and Walker2d environments across the four D4RL dataset types (Random, Medium, Medium-Replay, Medium-Expert), under the **Standard Challenge Configurations** specified in Table 4. The specific challenges required for evaluation depend on the algorithm category: **Online RL** baselines should be evaluated on Challenges 1, 2, and 3; **Offline RL** baselines on Challenge 4; and **Hybrid RL** algorithms on all four challenges. While this core protocol establishes a standardized high-difficulty baseline, we strongly encourage practitioners to also report results on lower-intensity settings (e.g., $\sigma_{low}$) and utilize the modular design of the B4MRL codebase to explore additional configurations for comprehensive diagnostic analysis.

## C    BENCHMARK IMPLEMENTATION DETAILS

In this section we provide further details regarding our benchmarks, and discuss how different benchmarks could be customized using B4MRL. Each of our hybrid RL benchmarks consists of two components: (1) an imperfect simulator with sim2real error, and (2) an offline dataset with a offline2real error. Motivation, explanation and examples for these errors are discussed thoroughly in Section 3.

We now provide a list of possible sim2real errors that can be used in any MuJoCo environment, a list of offline2real that can be introduced to the D4RL MuJoCo datasets, and a list of new datasets that have offline2real errors. For the offline2real errors, we chose the same parameters we used for

Table 4: The B4MRL Core Evaluation Protocol. To ensure standardized comparison, we define the specific simulator + dataset pairings that constitute the core testbed. Future work should report results on these configurations across all 3 environments (HalfCheetah, Hopper, Walker2d) and 4 D4RL dataset types (Random, Medium, Medium-Replay, Medium-Expert).

| Challenge ID | Name | Target Algorithms | Simulator Configuration | Dataset Configuration |
|---|---|---|---|---|
| **Challenge 1** | Modeling Error | Online, Hybrid | **High Intensity** (Env-Specific: gravity, friction, or leg size) | Standard D4RL |
| **Challenge 2** | Partial Obs. | Online, Hybrid | **Hidden Dim** ($h_{high}$) | Standard D4RL |
| | | | **Obs. Noise** ($\sigma_{high}$) | Standard D4RL |
| **Challenge 3** | Action Disc. | Online, Hybrid | **Action Noise** ($\sigma_{high}$) | Standard D4RL |
| **Challenge 4** | Confounding | Offline, Hybrid | **Ch 1 Config** (Modeling Error) | **Conf. Hidden Dim** ($h_{high}$) |
| | | | | **Conf. Noise** ($\sigma_{high}$) |

sim2real, in order to be able to compare. In addition we provide information regarding the Highway environment, where the agent's goal is to drive fast enough and avoid collisions on a multi-lane road.

**Sim2Real.** We chose the specific parameters for each type of error according to how well did SAC perform on that simulation, aiming to provide two levels of errors per category. A list and details of all sim2real errors (summarised in Table 5):

- **Transition error (challenge 1 – modelling error)**: For MuJoCo environments, we create a simulator with transition error by modifying the environment's XML file provided by the gym package. Additional simulators can be easily created by adding new modified XML files to the relevant directory. For the Highway environment, the modelling error is the difference between the amount of vehicles on the road during train and during test. This difference could make the agent learn a more cautious policy in order to avoid collisions, but when the road is free it might not achieve optimal reward.

- **Observation noise (challenge 2 – partial observability):** The environment's dynamics are unchanged, but we add Gaussian noise to the observation, and return only the noisy observation to the user. We denote the two noise levels by their standard deviation $\sigma_{low}$ for low added noise, and $\sigma_{high}$ for high added noise.

- **Hidden variables (challenge 2 – partial observability):** The environment's dynamics are unchanged, but we fix a specific observation dimension to zero before returning it to the user. The reason we zero the dimension and not remove it entirely is that we do not want to change the observation-space definition of the environment. This should make implementing hybrid algorithms easier, since using two sources of data (simulator and dataset) with different dimensionality of variables might result in two different observation-spaces. We denote the two choices of hidden dimension by $h_{low}$ for low effect and $h_{high}$ for high effect.

- **Action noise (challenge 3 – action discrepancy):** When a user selects a specific action and sends it to the simulator (e.g., via the `step` method), we modify the action by adding Gaussian noise.

**Offline2Real.** For each type of error, we used the same parameters as in the sim2real errors (as summarised in Table 5). A list and details of all offline2real errors:

- **Observation noise (challenge 4 – hidden confounders):** We go over all the observations in the D4RL dataset, and add Gaussian noise to each observation, and for each observation dimension. That means that a value sampled from a zero mean unit variance Gaussian distribution is added independently to each entry in the observation matrix. Note that we sample the noise matrix once per dataset, and run all experiments on the same noisy dataset (e.g., a different noise matrix is used for HalfCheetah-medium, for HalfCheetah-medium-expert, and for every other dataset). To obtain different noise levels, we multiply the sampled noise by a magnitude scalar $\sigma_{low}$ or $\sigma_{high}$, using the same values as in the sim2real. We

Table 5: Sim2real errors on online simulatros. $g$ is the gravitational parameter, $f$ is the friction parameter, 'leg' is the leg length, $v_x, v_y$ are the velocities, $y$ is the position in the $y$ axis. 'cars' signify the amount of cars on the road for the highway environment, where in test time there were only only 3 cars on the road. The partial observability row refers to variables that were hidden from the agent by the simulator.

| Error type | Amount | HalfCheetah | Hopper | Walker2D | Highway |
|---|---|---|---|---|---|
| Transition error | - | $g_{\text{sim}} = 2g$ 
 $f_{\text{sim}} = 0.3f$ | $g_{\text{sim}} = 2g$ 
 $\text{leg}_{\text{sim}} = 0.8$ | $g_{\text{sim}} = 2g$ 
 $f_{\text{sim}} = 0.3f$ | $\#\text{cars}_{\text{sim}} = 15$ |
| Observation noise | $\sigma_{\text{low}}$ | 0.01 | 0.001 | 0.01 | 0.5 |
|  | $\sigma_{\text{high}}$ | 0.05 | 0.02 | 0.03 | 1.0 |
| Partial observability | $h_{\text{low}}$ | #16 | #10 | #16 | $(v_x, v_y)$ |
|  | $h_{\text{high}}$ | #9 | #9 | #9 | $(y, v_x, v_y)$ |
| Action noise | $\sigma_{\text{low}}$ | 0.2 | 0.5 | 0.2 | 0.5 |
|  | $\sigma_{\text{high}}$ | 0.5 | 1.0 | 0.5 | 1.0 |

provide the noise matrices and code for generating the noise, so that users can experiment with more settings as well.

- **Hidden variables (challenge 4 – hidden confounders):** We go over all the observations in a given dataset, and zero the chosen dimensions dimension. A user can select any dimension they wish to zero (or a list of them), and any dataset. We chose for our benchmark the same dimensions used in the sim2real hidden-variable benchmark. Note that in this case, the agent that generated the data saw the hidden variable when making its decisions.

In addition to the modified D4RL datasets, we also provide new complementary datasets, that were generated by an agent trained on a noisy environment. The agent was trained using the SAC algorithm provided by the `stable-baselines` package on the simulators listed below. Here too we used the same parameters as sim2real (as shown in Table 5). We provide the datasets, the agent that made the datasets, and code to recreate the agent, so that users can follow the same process and create new agents on different noisy environments.

- **Hidden variables** Similarly, to the sim2real hidden-variables error, the agent was trained on a simulator with a zeroed variable. To generate the data, we used the agent that scored as close as possible to the medium dataset in D4RL. For example, in HalfCheetah we selected the agent that scored as close to a normalized score of 40 as possible.

- **Action delay** The agent was trained on the action-delay environment described in sim2real, and was used to collect a dataset of trajectories. Note that in this case, unlike the dataset mentioned above and the D4RL datasets, we collected a dataset of full trajectories, and not tuples of a single step in time.

**Combining sim2real and offline2real for Hybrid-RL algorithms.** The above sim2real and offline2real environments can be easily combined to form any hybrid-RL benchmark desired. We propose a representative set of benchmarks that cover most aspects discussed throughout the paper. For results we refer the readers to Section G.

## D    BASELINE IMPLEMENTATION DETAILS

In this section we provide further details on the baselines used for the experiments in Section 4, including more information for HyMOPO.

Table 6: SAC results on the different environments with the parameters described in Table 5. Results on MuJoCo environemnts are normalized as in D4RL. Results on the Highway enviroment are not normalized (range of the reward is approximately between 0 and 22).

| Error type | HalfCheetah | Hopper | Walker2D | Highway |
|---|---|---|---|---|
| Transition error | $65.9 \pm 11.7$ | $67.3 \pm 35.1$ | $50.6 \pm 5.5$ | $12.5 \pm 5.7$ |
| | $35.1 \pm 2.4$ | $32.3 \pm 11.5$ | $70.6 \pm 17.5$ | |
| Observation noise | $80.0 \pm 1.4$ | $81.3 \pm 23.2$ | $81.8 \pm 7.5$ | $19.4 \pm 4.5$ |
| | $45.7 \pm 1.8$ | $59.7 \pm 24.7$ | $46.6 \pm 9.4$ | $18.4 \pm 6.3$ |
| Partial observability | $83.3 \pm 3.3$ | $77.5 \pm 13.7$ | $83.5 \pm 5.2$ | $18.6 \pm 4.9$ |
| | $64.0 \pm 1.1$ | $55.1 \pm 33.8$ | $51.1 \pm 29.4$ | $10.5 \pm 7.9$ |
| Action noise | $82.4 \pm 4.2$ | $82.3 \pm 19.5$ | $91.2 \pm 2.2$ | $17.9 \pm 2.8$ |
| | $62.7 \pm 0.5$ | $44.9 \pm 22.4$ | $63.9 \pm 29.2$ | $9.7 \pm 1.9$ |

In the paper we implement, or use prior implementations of two online-RL algorithms: TD3[1], and SAC[2], three offline-RL algorithms: TD3-BC[3], MOPO[4] and IQL[5], and three hybrid-RL algorithms: H2O[6], PAR-BC[7] and HyMOPO. For each algorithm we used the hyperparameters used in the respective paper. We argue that the errors discussed in this paper are not known in advance to the algorithm, therefore, searching for the hyperparameters that obtain the best reward on the real world environment is not reasonable. For HyMOPO, we used the same hyperparameters as in MOPO, with $\lambda = 0.0$, and $h = 5$ on HalfCheetah.

The HyMOPO algorithm is a model-based, dynamics-aware, policy optimization algorithm. In this approach, we first train a correction function $f$ that learns to fix the discrepancy between observed data and the simulator's outputs: This is done by running each observation-action tuple $(o_i, a_i)$ through the simulator and collecting its outputs, i.e., the simulator's computed next observations $o'_{i,\text{sim}}$. The correction function's goal is to learn an additive function that fixes the gap between the simulator's next observation and the next observation registered in the dataset. This is done by minimizing the following loss: $\mathcal{L}(f) = \frac{1}{N} \sum_i \|o'_i - (o'_{i,\text{sim}} + f(o_i, a_i))\|$, where $N$ is the number of observations in the dataset. We note that in the worst case, when the simulator is completely incorrect, the correcting function should learn to output $o'_{i,\text{sim}} - o'$, which would typically be as difficult as learning the transition function directly from data (i.e., learning a model that outputs the next state given the current state and action) as seen in other offline-RL algorithms such as MOPO.

To train the agent, we first initialize the state by randomly selecting one from the given dataset. Then the transition function, which consists of a simulator $T_{\text{sim}}$ and a correction function $f_\theta$, is used to determine the next observations up to a predetermined horizon. Finally, the reward is penalized by the amount of uncertainty in the transition model evaluations. These last steps are similar to the MOPO algorithm, with the difference that HyMOPO can combine the given dataset with a given simulator. In Algorithm 1 we provide full algorithmic description of HyMOPO. Parts of our algorithm are similar

---

[1]Code available at `https://github.com/sfujim/TD3`

[2]Stable-baselines implementation `https://stable-baselines3.readthedocs.io/`

[3]Code available at `https://github.com/sfujim/TD3_BC`

[4]Code for the original paper avialable at `https://github.com/tianheyu927/mopo`. However, we used a different implementation that is simpler to use and achieves the same results, found at `https://github.com/junming-yang/mopo`

[5]Code for the original paper available at `https://github.com/ikostrikov/implicit_q_learning/`. We used the pytorch version which achieves the same results at `https://https://github.com/Manchery/iql-pytorch/`

[6]Code available at `https://github.com/t6-thu/H2O`

[7]The code for the BC version of the PAR algorithm is available at `https://github.com/OffDynamicsRL/off-dynamics-rl`

to MBPO (Janner et al., 2019) and MOPO, with the modifications needed for becoming a hybrid-RL algorithm that can use a simulator together with a given dataset.

---

**Algorithm 1** HyMOPO algorithm for hybrid-RL

---

**Input:** offline dataset $\mathcal{D}$, simulator $T_{\text{sim}}$, an ensemble of $N$ learnable correction functions $\{f_\theta^i\}_{i=1}^N$, reward penalty coefficient $\lambda$, rollout horizon $h$, rollout batch size $b$.
**Init:** random weights $\theta$ for each in $f_\theta$ from $1...N$.
Evaluate $o'_{\text{sim}} = T_{\text{sim}}(o, a)$, for each tuple in $\mathcal{D}$ and add to $\mathcal{D}$
**for** each correction function $f_\theta^i$ in $i = 1...N$ **do**
    Train a probabilistic correction function on $\mathcal{D}$ batches:
    $f_\theta^i(o, a, o'_{\text{sim}}) = o'_{\text{sim}} + \mathcal{N}(\mu^i(o, a), \Sigma^i(o, a))$
**end for**
Initialize policy $\pi$ and empty replay buffer $\mathcal{D}_{\text{model}}$.
**for** epoch $1, 2, ...$ **do**
    Sample initial rollout state $o_1$ from $\mathcal{D}$
    **for** j=1,2,...,h **do**
        Sample an action $a_j \sim \pi(o_j)$
        Evaluate $o'_{\text{sim}} = T_{\text{sim}}(o_j, a_j)$
        Randomly select $f_\theta^i$ and sample an observation correction and reward $(\Delta o', r_j) \sim f_\theta^i(o_j, a_j)$
        Evaluate next state $o_{j+1} = o'_{\text{sim}} + \Delta o'$
        Evaluate penalized reward $\tilde{r}_j = r_j - \lambda \max_{i=1}^N \|\Sigma^i(o_j, a_j)\|_F$
        Add tuple $(o_j, a_j, \tilde{r}_j, o_{j+1})$ to $\mathcal{D}_{\text{model}}$
    **end for**
    Draw samples from $\mathcal{D} \cup \mathcal{D}_{\text{model}}$ to update $\pi$ using SAC
**end for**

---

# E  LIMITATIONS

While B4MRL introduces a comprehensive set of challenges for hybrid RL, it does not exhaustively cover all possible combinations of discrepancies and domains. We focus on representative and practically meaningful scenarios. To this end, we use synthetic datasets, which are well-known and widely used in the community (mainly based on D4RL Fu et al. (2020)), and which are also easily configurable in order to control and isolate different types of discrepancies – for example, adding confounding errors to the data. Although real-world data would provide the most realistic testbeds, separating the effects of confounding errors and other challenges, and cleanly comparing multiple online, offline and hybrid methods would be much more difficult. The benchmarks are designed to be modular and easy to use, enabling the community to explore additional combinations beyond those evaluated here (more details on benchmark implementation in Section C). We hope this paper serves as a starting point for future research in hybrid RL and that B4MRL provides a shared foundation for evaluating new algorithms under realistic conditions.

# F  BROADER IMPACT

While our benchmarks provide valuable tools for evaluating and improving offline, online, and hybrid RL algorithms, it is important to recognize their limitations. Strong performance on these controlled benchmarks does not guarantee reliable results across all real-world scenarios, which may feature more complex simulator discrepancies, dataset biases, and confounding factors. Our benchmarks are designed to foster transparency and robustness in RL research by enabling systematic testing of key challenges, but they represent only one step in the broader process of developing and validating RL methods. We encourage researchers and practitioners to interpret benchmark results carefully and conduct thorough testing in diverse real-world environments before deploying RL algorithms in critical applications. Additionally, while our work aims to advance safer and more reliable RL systems that could benefit domains such as robotics and healthcare, there remain potential risks if these technologies are misused or deployed without adequate safeguards. By understanding both the capabilities and limits of our benchmarks, the community can better drive responsible innovation in reinforcement learning.

## G  EXPERIMENTS

In this section we provide extra experimental results on all baselines and benchmarks for the `MuJoCo-HalfCheetah` environment, complementing the results displayed in Section 4. Moving forward, we see B4MRL as a dynamic benchmark that will develop and expand with new datasets and new tasks to evaluate the four challenges. Full implementations, datasets and more results will be made available on the project page on GitHub, which is available here `https://anonymous.4open.science/r/B4MRL-D8DD`. However, the code is also provided in the supplementary material. As for compute, we utilized a single NVIDIA A40 GPU for each of the experiments. We divide the results by the four challenges described in Section 3, on the benchmarks described in Section C.

**Modelling error.**   In Table 7, in Table 13, and in Table 15, we provide results on the HalfCheetah, the Walker2D, and the Hopper environments respectively, for the modelling-error challenge (challenge 1), which is modeled by changing one of the simulator's parameters in charge of the dynamics. In this experiment we observe that the friction discrepancy had a smaller effect on the ability of the online-RL algorithms to achieve higher rewards, when compared to the gravity modeling error. We also observe that in HalfCheetah, HyMOPO achieves higher rewards when compared to all other baselines, except when using random dataset with friction discrepancy, suggesting that in this setting HyMOPO is able to effectively use the information from both online and offline sources. We stress that HyMOPO is not suitable for the current MuJoCo implementation of Walker-2D and Hopper because the environment clips the observations to the range of $[-10, 10]$. In the correction function learning phase, HyMOPO takes an observation from the offline dataset, and uses the simulator to evaluate the next observation. However, the observations in the offline dataset are already clipped in the data gathering process, so when HyMOPO queries the simulator for the next observation, it returns an observation that is far from the true next observation. This discrepancy introduces significant noise into the simulator, which we found severely degrades the performance of HyMOPO.

**Partial observability.**   In Table 8, in Table 16, and in Table 12, we provide results on the HalfCheetah, the Walker2D, and the Hopper environments respectively for the partial-observability challenge (challenge 2), which is modeled by either removing a variable from the simulator's observation, or by adding Gaussian noise to the simulator's observations. Similarly to the modelling-error experiment, in the HalfCheatah environment, HyMOPO obtains better results than other baselines in most cases. However, as also discussed in Section 4, we see that in some cases it is better to use a single source of information than both. For instance, on the medium dataset, with $h_{\text{low}}$ discrepancy, SAC achieves mean reward of $83.3 \pm 3.3$ on the imperfect simulator, and MOPO achieves reward of $66.1 \pm 0.3$ on the medium offline dataset. Notably, both hybrid-RL algorithms are inferior to both MOPO and SAC, suggesting that combining sources of information does not guarantee results that are better than both.

In Table 9 we provide additional results on datasets we created that were generated by an agent that only has access to partial observations, which is modeled by removing variables from the observations. For the simulator, we used a simulator with gravity transition error. These results suggest that removing variables from the dataset has a stronger effect on performance compared to removing those same variables from the simulators.

**Confounding error**   In Table 11, in Table 14, and in Table 17, we provide results on the HalfCheetah, the Walker2D, and the Hopper environments respectively for the confounding error challenge (challenge 4), which is modeled similarly to the partial observability challenge, by either removing observations from the dataset or by adding Gaussian noise the the entire dataset observations. creating a discrepancy between what the agent generating the dataset used and what the offline method can use. For the simulator, in all environments, we used a simulator with gravity transition error. Continuing the discussion from Section 4, and addressing the added experiments we provide here, we observe the same phenomenon, where hidden confounding can have a very strong negative impact on the results.

Table 7: Results on HalfCheetah with modeling error (challenge 1).

| | | Online-RL (sim) | | Offline-RL (data) | | | Hybrid-RL | | |
|---|---|---|---|---|---|---|---|---|---|
| Dataset | Discrepancy | TD3 | SAC | MOPO | TD3-BC | IQL | H2O | PAR-BC | HyMOPO |
| Random | Friction | $45.0 \pm 9.5$ | $65.9 \pm 11.7$ | $36.2 \pm 0.9$ | $12.2 \pm 0.5$ | $14.7 \pm 3.2$ | $30.4 \pm 12.2$ | $36.5 \pm 15.0$ | $40.0 \pm 2.0$ |
| | Gravity | $35.3 \pm 1.9$ | $35.1 \pm 2.4$ | | | | $36.7 \pm 2.4$ | $36.9 \pm 2.2$ | $36.1 \pm 2.5$ |
| Medium | Friction | $45.0 \pm 9.5$ | $65.9 \pm 11.7$ | $66.1 \pm 0.3$ | $48.3 \pm 0.1$ | $48.5 \pm 0.4$ | $55.7 \pm 5.9$ | $78.1 \pm 9.2$ | $73.9 \pm 0.4$ |
| | Gravity | $35.3 \pm 1.9$ | $35.1 \pm 2.4$ | | | | $51.0 \pm 10.4$ | $43.7 \pm 1.2$ | $72.9 \pm 0.8$ |
| Medium replay | Friction | $45.0 \pm 9.5$ | $65.9 \pm 11.7$ | $67.8 \pm 2.4$ | $42.8 \pm 2.9$ | $44.4 \pm 0.1$ | $49.8 \pm 3.6$ | $39.7 \pm 11.1$ | $68.6 \pm 1.4$ |
| | Gravity | $35.3 \pm 1.9$ | $35.1 \pm 2.4$ | | | | $54.8 \pm 3.3$ | $40.7 \pm 0.9$ | $66.5 \pm 0.6$ |
| Medium expert | Friction | $45.0 \pm 9.5$ | $65.9 \pm 11.7$ | $49.2 \pm 14.5$ | $84.3 \pm 5.2$ | $94.3 \pm 0.3$ | $18.9 \pm 1.8$ | $95.2 \pm 0.5$ | $99.2 \pm 5.1$ |
| | Gravity | $35.3 \pm 1.9$ | $35.1 \pm 2.4$ | | | | $41.2 \pm 2.9$ | $89.8 \pm 1.6$ | $95.1 \pm 2.0$ |

Table 8: Results on HalfCheetah with partial observations (challenge 2).

| | | Online-RL (sim) | | Offline-RL (data) | | | Hybrid-RL | | |
|---|---|---|---|---|---|---|---|---|---|
| Dataset | Discrepancy | TD3 | SAC | MOPO | TD3-BC | IQL | H2O | PAR-BC | HyMOPO |
| Random | $\sigma_{low}$ | $75.1 \pm 12.4$ | $80.0 \pm 1.4$ | $36.2 \pm 0.9$ | $12.2 \pm 0.5$ | $14.7 \pm 3.2$ | $44.4 \pm 12.2$ | $38.6 \pm 29.5$ | $37.8 \pm 2.8$ |
| | $\sigma_{high}$ | $49.0 \pm 1.9$ | $45.7 \pm 1.8$ | | | | $33.0 \pm 5.3$ | $25.0 \pm 16.1$ | $33.2 \pm 3.1$ |
| | $h_{low}$ | $47.8 \pm 6.5$ | $83.3 \pm 3.3$ | | | | $25.2 \pm 2.4$ | $35.7 \pm 1.0$ | $35.7 \pm 1.0$ |
| | $h_{high}$ | $62.8 \pm 2.5$ | $64.0 \pm 1.1$ | | | | $21.9 \pm 1.2$ | $39.6 \pm 0.2$ | $39.6 \pm 0.2$ |
| Medium | $\sigma_{low}$ | $75.1 \pm 12.4$ | $80.0 \pm 1.4$ | $66.1 \pm 0.3$ | $48.3 \pm 0.1$ | $48.5 \pm 0.4$ | $60.1 \pm 1.4$ | $76.4 \pm 1.0$ | $76.8 \pm 0.6$ |
| | $\sigma_{high}$ | $49.0 \pm 1.9$ | $45.7 \pm 1.8$ | | | | $54.0 \pm 0.2$ | $46.4 \pm 0.0$ | $55.9 \pm 1.0$ |
| | $h_{low}$ | $47.8 \pm 6.5$ | $83.3 \pm 3.3$ | | | | $51.7 \pm 8.8$ | $76.1 \pm 2.0$ | $76.1 \pm 2.0$ |
| | $h_{high}$ | $62.8 \pm 2.5$ | $64.0 \pm 1.1$ | | | | $44.0 \pm 2.0$ | $52.5 \pm 6.7$ | $52.5 \pm 6.7$ |
| Medium replay | $\sigma_{low}$ | $75.1 \pm 12.4$ | $80.0 \pm 1.4$ | $67.8 \pm 2.4$ | $42.8 \pm 2.9$ | $44.4 \pm 0.1$ | $53.8 \pm 2.8$ | $70.4 \pm 3.0$ | $73.8 \pm 2.2$ |
| | $\sigma_{high}$ | $49.0 \pm 1.9$ | $45.7 \pm 1.8$ | | | | $50.1 \pm 0.4$ | $45.1 \pm 0.5$ | $54.8 \pm 0.4$ |
| | $h_{low}$ | $47.8 \pm 6.5$ | $83.3 \pm 3.3$ | | | | $53.6 \pm 0.8$ | $66.0 \pm 3.8$ | $66.0 \pm 3.8$ |
| | $h_{high}$ | $62.8 \pm 2.5$ | $64.0 \pm 1.1$ | | | | $46.2 \pm 0.8$ | $65.4 \pm 4.3$ | $65.4 \pm 4.3$ |
| Medium expert | $\sigma_{low}$ | $75.1 \pm 12.4$ | $80.0 \pm 1.4$ | $49.2 \pm 14.5$ | $84.3 \pm 5.2$ | $94.3 \pm 0.3$ | $44.8 \pm 9.5$ | $95.1 \pm 0.3$ | $101.6 \pm 0.3$ |
| | $\sigma_{high}$ | $49.0 \pm 1.9$ | $45.7 \pm 1.8$ | | | | $33.9 \pm 0.3$ | $77.2 \pm 5.9$ | $56.2 \pm 0.7$ |
| | $h_{low}$ | $47.8 \pm 6.5$ | $83.3 \pm 3.3$ | | | | $47.8 \pm 4.5$ | $101.6 \pm 1.0$ | $101.6 \pm 1.0$ |
| | $h_{high}$ | $62.8 \pm 2.5$ | $64.0 \pm 1.1$ | | | | $33.8 \pm 5.6$ | $79.9 \pm 7.0$ | $79.9 \pm 7.0$ |

Table 9: Results on HalfCheetah on datasets with partial observations, but without confounding (challenge 2). Online and hybrid RL models have access to a simulator with modeling error as well.

| | Online-RL (sim) | | Offline-RL (data) | | | Hybrid-RL | |
|---|---|---|---|---|---|---|---|
| Dataset | TD3 | SAC | MOPO | TD3-BC | IQL | H2O | HyMOPO |
| $h_{low}$ | $35.3 \pm 1.9$ | $35.1 \pm 2.4$ | $55.1 \pm 0.4$ | $45.9 \pm 0.1$ | $47.5 \pm 0.1$ | $45.8 \pm 5.1$ | $72.4 \pm 1.5$ |
| $h_{low}$-history | | | $56.5 \pm 0.8$ | $51.7 \pm 0.3$ | $52.1 \pm 0.2$ | $52.5 \pm 0.8$ | $71.2 \pm 1.3$ |
| $h_{high}$ | | | $0.7 \pm 0.4$ | $50.4 \pm 0.5$ | $48.1 \pm 0.1$ | $14.1 \pm 11.2$ | $37.1 \pm 1.4$ |
| $h_{high}$-history | | | $3.0 \pm 1.1$ | $49.8 \pm 0.3$ | $48.1 \pm 0.2$ | $48.5 \pm 1.8$ | $32.0 \pm 1.5$ |

Table 10: Results on HalfCheetah with action error (challenge 3).

| Dataset | Discrepancy | Online-RL (sim) | | Offline-RL (data) | | | Hybrid-RL | | |
|---|---|---|---|---|---|---|---|---|---|
| | | TD3 | SAC | MOPO | TD3-BC | IQL | H2O | PAR-BC | HyMOPO |
| Random | $\sigma_{low}$ | $78.7 \pm 9.1$ | $82.4 \pm 4.2$ | $36.2 \pm 0.9$ | $12.2 \pm 0.5$ | $14.7 \pm 3.2$ | $32.7 \pm 2.4$ | $37.1 \pm 27.6$ | $41.4 \pm 2.7$ |
| | $\sigma_{high}$ | $67.7 \pm 1.3$ | $62.7 \pm 0.5$ | | | | $23.5 \pm 1.6$ | $59.4 \pm 1.1$ | $36.9 \pm 1.9$ |
| Medium | $\sigma_{low}$ | $78.7 \pm 9.1$ | $82.4 \pm 4.2$ | $66.1 \pm 0.3$ | $48.3 \pm 0.1$ | $48.5 \pm 0.4$ | $57.7 \pm 0.5$ | $67.3 \pm 1.4$ | $75.4 \pm 2.9$ |
| | $\sigma_{high}$ | $67.7 \pm 1.3$ | $62.7 \pm 0.5$ | | | | $60.3 \pm 0.9$ | $47.5 \pm 0.3$ | $54.3 \pm 1.0$ |
| Medium replay | $\sigma_{low}$ | $78.7 \pm 9.1$ | $82.4 \pm 4.2$ | $67.8 \pm 2.4$ | $42.8 \pm 2.9$ | $44.4 \pm 0.1$ | $54.7 \pm 0.6$ | $60.0 \pm 1.3$ | $68.5 \pm 4.7$ |
| | $\sigma_{high}$ | $67.7 \pm 1.3$ | $62.7 \pm 0.5$ | | | | $58.0 \pm 1.7$ | $45.5 \pm 0.3$ | $48.0 \pm 0.2$ |
| Medium expert | $\sigma_{low}$ | $78.7 \pm 9.1$ | $82.4 \pm 4.2$ | $49.2 \pm 14.5$ | $84.3 \pm 5.2$ | $94.3 \pm 0.3$ | $36.0 \pm 3.8$ | $95.9 \pm 0.5$ | $89.0 \pm 2.8$ |
| | $\sigma_{high}$ | $67.7 \pm 1.3$ | $62.7 \pm 0.5$ | | | | $38.1 \pm 12.7$ | $90.0 \pm 3.7$ | $52.7 \pm 1.4$ |

Table 11: Normalized reward on HalfCheetah environment, on four types of datasets, all with confounding errors. Online and Hybrid models also have access to a simulator with a transition error in the gravitational parameter.

| Dataset | Confounding | Online-RL (sim) | | Offline-RL (data) | | | Hybrid-RL | | |
|---|---|---|---|---|---|---|---|---|---|
| | | TD3 | SAC | MOPO | TD3-BC | IQL | H2O | PAR-BC | HyMOPO |
| Random | $\sigma_{low}$ | $35.3 \pm 1.9$ | $35.1 \pm 2.4$ | $36.6 \pm 2.6$ | $11.4 \pm 1.8$ | $11.7 \pm 3.5$ | $31.0 \pm 1.0$ | $37.8 \pm 2.4$ | $38.3 \pm 1.8$ |
| | $\sigma_{high}$ | | | $24.1 \pm 1.4$ | $10.3 \pm 0.8$ | $2.6 \pm 0.1$ | $30.1 \pm 2.1$ | $9.6 \pm 0.7$ | $33.6 \pm 1.6$ |
| | $h_{low}$ | | | $37.4 \pm 1.0$ | $11.7 \pm 0.6$ | $12.4 \pm 3.0$ | $34.2 \pm 1.1$ | $2.2 \pm 0.0$ | $31.5 \pm 3.3$ |
| | $h_{high}$ | | | $26.2 \pm 3.3$ | $9.0 \pm 0.9$ | $6.6 \pm 3.0$ | $31.0 \pm 2.1$ | $2.2 \pm 0.0$ | $29.9 \pm 0.6$ |
| Medium | $\sigma_{low}$ | $35.3 \pm 1.9$ | $35.1 \pm 2.4$ | $29.6 \pm 13.8$ | $47.5 \pm 0.5$ | $48.4 \pm 0.2$ | $42.5 \pm 7.2$ | $42.4 \pm 1.0$ | $74.9 \pm 3.9$ |
| | $\sigma_{high}$ | | | $-0.1 \pm 0.7$ | $41.0 \pm 0.7$ | $37.1 \pm 2.1$ | $17.3 \pm 7.0$ | $36.4 \pm 0.8$ | $9.8 \pm 2.6$ |
| | $h_{low}$ | | | $60.6 \pm 7.1$ | $48.2 \pm 0.2$ | $48.4 \pm 0.2$ | $54.3 \pm 2.5$ | $44.8 \pm 0.8$ | $73.4 \pm 0.9$ |
| | $h_{high}$ | | | $29.4 \pm 4.1$ | $46.1 \pm 0.5$ | $46.5 \pm 0.1$ | $34.5 \pm 3.4$ | $42.3 \pm 0.4$ | $35.2 \pm 1.7$ |
| Medium replay | $\sigma_{low}$ | $35.3 \pm 1.9$ | $35.1 \pm 2.4$ | $53.6 \pm 5.6$ | $44.4 \pm 0.4$ | $44.3 \pm 0.0$ | $47.0 \pm 8.8$ | $39.9 \pm 0.2$ | $73.2 \pm 1.2$ |
| | $\sigma_{high}$ | | | $14.7 \pm 4.5$ | $38.4 \pm 1.4$ | $35.3 \pm 3.3$ | $21.8 \pm 4.4$ | $37.3 \pm 0.5$ | $38.9 \pm 0.4$ |
| | $h_{low}$ | | | $58.7 \pm 8.0$ | $44.6 \pm 0.3$ | $43.8 \pm 1.1$ | $49.9 \pm 4.9$ | $43.1 \pm 0.4$ | $65.2 \pm 0.4$ |
| | $h_{high}$ | | | $32.9 \pm 1.1$ | $41.6 \pm 1.9$ | $42.5 \pm 0.0$ | $22.7 \pm 7.5$ | $38.3 \pm 2.3$ | $37.1 \pm 1.4$ |
| Medium expert | $\sigma_{low}$ | $35.3 \pm 1.9$ | $35.1 \pm 2.4$ | $-0.1 \pm 0.6$ | $78.6 \pm 4.3$ | $67.2 \pm 6.4$ | $34.6 \pm 3.4$ | $66.1 \pm 2.7$ | $80.7 \pm 5.8$ |
| | $\sigma_{high}$ | | | $-1.0 \pm 1.1$ | $33.5 \pm 2.8$ | $28.3 \pm 5.7$ | $13.0 \pm 9.1$ | $35.4 \pm 2.0$ | $16.8 \pm 3.0$ |
| | $h_{low}$ | | | $52.7 \pm 4.4$ | $91.4 \pm 2.0$ | $90.7 \pm 3.0$ | $34.3 \pm 7.7$ | $92.5 \pm 2.1$ | $99.2 \pm 0.8$ |
| | $h_{high}$ | | | $2.9 \pm 0.8$ | $74.3 \pm 4.1$ | $64.0 \pm 3.4$ | $18.7 \pm 4.7$ | $60.0 \pm 1.8$ | $27.0 \pm 2.0$ |

Table 12: Results on Walker2D with partial observations (challenge 2).

| Dataset | Discrepancy | Online-RL (sim) | | Offline-RL (data) | | | Hybrid-RL |
|---|---|---|---|---|---|---|---|
| | | TD3 | SAC | MOPO | TD3-BC | IQL | H2O |
| Random | $\sigma_{low}$ | 73.2 ± 13.8 | 81.8 ± 6.2 | 3.1 ± 2.2 | 3.4 ± 1.9 | 4.9 ± 1.7 | 10.6 ± 2.7 |
| | $\sigma_{high}$ | 30.4 ± 12.7 | 46.6 ± 7.7 | | | | 12.6 ± 3.7 |
| | $h_{low}$ | 21.9 ± 12.4 | 83.6 ± 4.2 | | | | 7.8 ± 2.3 |
| | $h_{high}$ | 45.5 ± 25.2 | 51.1 ± 24.0 | | | | 6.0 ± 4.8 |
| Medium | $\sigma_{low}$ | 73.2 ± 13.8 | 81.8 ± 6.2 | -0.1 ± 0.0 | 83.7 ± 3.1 | 75.4 ± 4.3 | 40.1 ± 9.7 |
| | $\sigma_{high}$ | 30.4 ± 12.7 | 46.6 ± 7.7 | | | | 20.3 ± 11.3 |
| | $h_{low}$ | 21.9 ± 12.4 | 83.6 ± 4.2 | | | | 25.7 ± 5.4 |
| | $h_{high}$ | 45.5 ± 25.2 | 51.1 ± 24.0 | | | | 16.7 ± 16.9 |
| Medium replay | $\sigma_{low}$ | 73.2 ± 13.8 | 81.8 ± 6.2 | 72.1 ± 13.3 | 83.5 ± 0.9 | 81.2 ± 2.7 | 70.3 ± 17.0 |
| | $\sigma_{high}$ | 30.4 ± 12.7 | 46.6 ± 7.7 | | | | 69.2 ± 18.7 |
| | $h_{low}$ | 21.9 ± 12.4 | 83.6 ± 4.2 | | | | 32.7 ± 14.4 |
| | $h_{high}$ | 45.5 ± 25.2 | 51.1 ± 24.0 | | | | 16.4 ± 4.8 |
| Medium expert | $\sigma_{low}$ | 73.2 ± 13.8 | 81.8 ± 6.2 | 24.9 ± 27.7 | 110.0 ± 0.1 | 112.0 ± 0.3 | 39.5 ± 36.7 |
| | $\sigma_{high}$ | 30.4 ± 12.7 | 46.6 ± 7.7 | | | | 22.6 ± 7.4 |
| | $h_{low}$ | 21.9 ± 12.4 | 83.6 ± 4.2 | | | | 70.6 ± 19.6 |
| | $h_{high}$ | 45.5 ± 25.2 | 51.1 ± 24.0 | | | | 22.3 ± 15.5 |

Table 13: Results on Walker2D with modeling error (challenge 1).

| Dataset | Discrepancy | Online-RL (sim) | | Offline-RL (data) | | | Hybrid-RL |
|---|---|---|---|---|---|---|---|
| | | TD3 | SAC | MOPO | TD3-BC | IQL | H2O |
| Random | Friction | 69.4 ± 7.9 | 71.9 ± 17.6 | 3.1 ± 2.2 | 3.4 ± 1.9 | 4.9 ± 1.7 | 7.4 ± 3.2 |
| | Gravity | 45.6 ± 20.2 | 50.7 ± 4.5 | | | | 12.3 ± 6.7 |
| Medium | Friction | 69.4 ± 7.9 | 71.9 ± 17.6 | -0.1 ± 0.0 | 83.7 ± 3.1 | 75.4 ± 4.3 | 31.5 ± 3.2 |
| | Gravity | 45.6 ± 20.2 | 50.7 ± 4.5 | | | | 41.1 ± 26.1 |
| Medium replay | Friction | 69.4 ± 7.9 | 71.9 ± 17.6 | 72.1 ± 13.3 | 83.5 ± 0.9 | 81.2 ± 2.7 | 82.5 ± 7.5 |
| | Gravity | 45.6 ± 20.2 | 50.7 ± 4.5 | | | | 48.7 ± 22.8 |
| Medium expert | Friction | 69.4 ± 7.9 | 71.9 ± 17.6 | 24.9 ± 27.7 | 110.0 ± 0.1 | 112.0 ± 0.3 | 37.0 ± 30.1 |
| | Gravity | 45.6 ± 20.2 | 50.7 ± 4.5 | | | | 40.7 ± 30.3 |

Table 14: Normalized reward on Walker2D environment, on four types of datasets, all with confounding errors. Online and Hybrid models also have access to a simulator with transition error in the gravitational parameter.

| Dataset | Conf. | Online-RL (sim) | | Offline-RL (data) | | | Hybrid-RL | |
|---|---|---|---|---|---|---|---|---|
| | | TD3 | SAC | MOPO | TD3-BC | IQL | H2O | PAR-BC |
| Random | $h_{low}$ | 45.6 ± 20.2 | 50.7 ± 4.5 | 6.7 ± 9.2 | 3.5 ± 1.9 | 4.7 ± 0.3 | 7.5 ± 3.2 | 24.5 ± 18.4 |
| | $h_{high}$ | | | 14.3 ± 9.1 | 2.6 ± 3.1 | 4.9 ± 0.2 | 4.9 ± 4.0 | 21.8 ± 0.0 |
| Medium | $h_{low}$ | 45.6 ± 20.2 | 50.7 ± 4.5 | -0.1 ± 0.0 | 84.5 ± 0.3 | 72.7 ± 6.4 | 25.5 ± 5.0 | 82.9 ± 2.0 |
| | $h_{high}$ | | | -0.1 ± 0.0 | 75.7 ± 3.4 | 72.7 ± 6.2 | 41.7 ± 20.3 | 79.4 ± 1.2 |
| Medium replay | $h_{low}$ | 45.6 ± 20.2 | 50.7 ± 4.5 | 39.1 ± 16.9 | 58.8 ± 37.5 | 68.6 ± 2.4 | 45.1 ± 13.5 | 66.3 ± 5.8 |
| | $h_{high}$ | | | 15.3 ± 0.8 | 7.3 ± 2.1 | 63.6 ± 6.4 | 19.3 ± 8.2 | 62.6 ± 14.0 |
| Medium expert | $h_{low}$ | 45.6 ± 20.2 | 50.7 ± 4.5 | 1.4 ± 2.2 | 109.7 ± 0.4 | 95.7 ± 23.0 | 16.6 ± 4.0 | 97.6 ± 8.1 |
| | $h_{high}$ | | | 8.7 ± 0.7 | 105.8 ± 6.0 | 102.9 ± 4.4 | 17.8 ± 3.1 | 91.0 ± 1.9 |

Table 15: Results on Hopper with modeling error (challenge 1).

| Dataset | Discrepancy | Online-RL (sim) | | Offline-RL (data) | | | Hybrid-RL | |
|---|---|---|---|---|---|---|---|---|
| | | TD3 | SAC | MOPO | TD3-BC | IQL | H2O | PAR-BC |
| Random | Leg | 67.4 ± 33.5 | 67.4 ± 28.7 | 11.2 ± 3.9 | 8.8 ± 0.5 | 7.5 ± 0.4 | 14.6 ± 5.5 | 62.0 ± 30.4 |
| | Gravity | 13.8 ± 4.9 | 10.4 ± 3.8 | | | | 19.5 ± 9.3 | 10.9 ± 1.9 |
| Medium | Leg | 67.4 ± 33.5 | 67.4 ± 28.7 | 34.5 ± 19.5 | 58.9 ± 1.9 | 62.6 ± 6.7 | 49.5 ± 36.5 | 28.5 ± 1.7 |
| | Gravity | 13.8 ± 4.9 | 10.4 ± 3.8 | | | | 7.9 ± 1.9 | 42.4 ± 3.1 |
| Medium replay | Leg | 67.4 ± 33.5 | 67.4 ± 28.7 | 27.6 ± 4.9 | 72.7 ± 28.9 | 84.7 ± 14.7 | 87.0 ± 11.9 | 55.1 ± 21.5 |
| | Gravity | 13.8 ± 4.9 | 10.4 ± 3.8 | | | | 58.0 ± 37.3 | 39.8 ± 25.4 |
| Medium expert | Leg | 67.4 ± 33.5 | 67.4 ± 28.7 | 19.9 ± 8.5 | 103.2 ± 9.5 | 88.5 ± 27.5 | 89.2 ± 11.4 | 99.3 ± 3.9 |
| | Gravity | 13.8 ± 4.9 | 10.4 ± 3.8 | | | | 19.6 ± 11.6 | 88.8 ± 4.3 |

Table 16: Results on Hopper with partial observations (challenge 2).

| Dataset | Discrepancy | Online-RL (sim) | | Offline-RL (data) | | | Hybrid-RL | |
|---|---|---|---|---|---|---|---|---|
| | | TD3 | SAC | MOPO | TD3-BC | IQL | H2O | PAR-BC |
| Random | $\sigma_{low}$ | 98.8 ± 2.6 | 81.3 ± 18.9 | 11.2 ± 3.9 | 8.8 ± 0.5 | 7.5 ± 0.4 | 67.8 ± 34.6 | 81.4 ± 28.5 |
| | $\sigma_{high}$ | 39.2 ± 24.8 | 37.0 ± 21.3 | | | | 20.8 ± 1.7 | 45.4 ± 34.7 |
| | $h_{low}$ | 7.2 ± 5.1 | 55.1 ± 27.6 | | | | 23.2 ± 6.3 | 30.7 ± 1.0 |
| | $h_{high}$ | 14.0 ± 15.4 | 77.5 ± 11.2 | | | | 4.8 ± 3.0 | 14.3 ± 6.0 |
| Medium | $\sigma_{low}$ | 98.8 ± 2.6 | 81.3 ± 18.9 | 34.5 ± 19.5 | 58.9 ± 1.9 | 62.6 ± 6.7 | 83.9 ± 6.8 | 75.3 ± 29.7 |
| | $\sigma_{high}$ | 39.2 ± 24.8 | 37.0 ± 21.3 | | | | 36.5 ± 5.6 | 18.8 ± 0.8 |
| | $h_{low}$ | 7.2 ± 5.1 | 55.1 ± 27.6 | | | | 38.1 ± 19.8 | 35.9 ± 6.5 |
| | $h_{high}$ | 14.0 ± 15.4 | 77.5 ± 11.2 | | | | 30.1 ± 2.8 | 25.2 ± 2.9 |
| Medium replay | $\sigma_{low}$ | 98.8 ± 2.6 | 81.3 ± 18.9 | 27.6 ± 4.9 | 72.7 ± 28.9 | 84.7 ± 14.7 | 74.1 ± 17.8 | 54.9 ± 31.6 |
| | $\sigma_{high}$ | 39.2 ± 24.8 | 37.0 ± 21.3 | | | | 69.5 ± 25.1 | 26.5 ± 2.9 |
| | $h_{low}$ | 7.2 ± 5.1 | 55.1 ± 27.6 | | | | 17.4 ± 10.9 | 18.1 ± 4.1 |
| | $h_{high}$ | 14.0 ± 15.4 | 77.5 ± 11.2 | | | | 33.2 ± 2.6 | 42.9 ± 12.2 |
| Medium expert | $\sigma_{low}$ | 98.8 ± 2.6 | 81.3 ± 18.9 | 19.9 ± 8.5 | 103.2 ± 9.5 | 88.5 ± 27.5 | 47.8 ± 3.5 | 34.9 ± 3.7 |
| | $\sigma_{high}$ | 39.2 ± 24.8 | 37.0 ± 21.3 | | | | 41.6 ± 7.5 | 18.9 ± 1.0 |
| | $h_{low}$ | 7.2 ± 5.1 | 55.1 ± 27.6 | | | | 27.8 ± 18.4 | 62.5 ± 31.9 |
| | $h_{high}$ | 14.0 ± 15.4 | 77.5 ± 11.2 | | | | 60.1 ± 23.5 | 52.5 ± 30.3 |

Table 17: Normalized reward on Hopper environment, on four types of datasets, all with confounding errors. Online and Hybrid models also have access to a simulator with transition error in the gravitational parameter.

| Dataset | Conf. | Online-RL (sim) | | Offline-RL (data) | | | Hybrid-RL | |
| | | TD3 | SAC | MOPO | TD3-BC | IQL | H2O | PAR-BC |
|---|---|---|---|---|---|---|---|---|
| Random | $h_{\text{low}}$ | $13.8 \pm 4.9$ | $10.4 \pm 3.8$ | $14.5 \pm 12.3$ | $8.5 \pm 0.2$ | $7.3 \pm 0.3$ | $23.3 \pm 9.1$ | $31.3 \pm 0.1$ |
| | $h_{high}$ | | | $26.1 \pm 7.8$ | $8.4 \pm 0.3$ | $7.2 \pm 0.3$ | $19.3 \pm 8.5$ | $31.5 \pm 0.0$ |
| Medium | $h_{\text{low}}$ | $13.8 \pm 4.9$ | $10.4 \pm 3.8$ | $24.3 \pm 11.5$ | $48.8 \pm 4.4$ | $51.0 \pm 2.5$ | $38.4 \pm 12.7$ | $45.1 \pm 4.3$ |
| | $h_{high}$ | | | $2.4 \pm 0.4$ | $55.9 \pm 2.6$ | $50.0 \pm 1.7$ | $13.3 \pm 10.8$ | $46.8 \pm 2.2$ |
| Medium replay | $h_{\text{low}}$ | $13.8 \pm 4.9$ | $10.4 \pm 3.8$ | $16.6 \pm 2.5$ | $59.0 \pm 27.9$ | $73.2 \pm 20.3$ | $26.3 \pm 11.0$ | $26.9 \pm 15.8$ |
| | $h_{high}$ | | | $15.9 \pm 2.3$ | $49.8 \pm 3.3$ | $54.0 \pm 11.9$ | $9.0 \pm 0.3$ | $41.2 \pm 20.7$ |
| Medium expert | $h_{\text{low}}$ | $13.8 \pm 4.9$ | $10.4 \pm 3.8$ | $24.3 \pm 9.0$ | $102.8 \pm 4.3$ | $54.2 \pm 35.8$ | $33.3 \pm 35.8$ | $75.7 \pm 12.0$ |
| | $h_{high}$ | | | $23.3 \pm 3.9$ | $51.9 \pm 13.5$ | $53.2 \pm 32.3$ | $20.8 \pm 9.0$ | $44.1 \pm 9.7$ |

