# OpenReview forum: "Benchmarks for Reinforcement Learning with Biased Offline Data and Imperfect Simulators"
_ICLR.cc/2026/Conference — Submitted to ICLR 2026_

### Official Review · Reviewer_5Jp9 · 2025-10-29

**Soundness:** 2
**Presentation:** 2
**Contribution:** 2
**Rating:** 2
**Confidence:** 4

**Summary:**

This work presents B4MRL, a set of offline datasets and code to add errors to MuJoCo and Highway simulators. The aim is to provide a benchmark for evaluating challenges in hybrid simulator-augmented offline RL, including modeling error, partial observability, state/action discrepancies, and hidden confounding.

**Strengths:**

The problem addressed by the paper is a relevant practical problem and the paper includes some interesting analysis.

**Weaknesses:**

**W1.** This work releases some datasets and code for modifying two simulators and presents results with some prior methods. I am not sure this is enough for a paper at this venue. Other reviewers may have a different opinion, but, since the aim of the paper is to aid future algorithm development, I think a paper of this kind should additionally include:

- (A) Specification of a small collection of experiments that serves as a standard testbed for developing future new methods (ie. similar to how D4RL specifies the -random, -mixed, -medium, etc.). Otherwise, there are so many degrees of freedom that future papers will all pick different settings, making fair evaluation of methods extremely difficult.

- (B) A more complete codebase benchmarking current methods on each of these experiments (e.g. potentially based on Unifloral or CORL). At the moment the baseline implementations are drawn from seven different codebases, which likely introduces confounding variables due to differences in implementation details beyond the core algorithms, which makes comparisons and conclusions difficult. I think for a paper of this kind, rigorous evaluation of prior methods should be part of the contribution.

**W2.** Since the main contribution of this paper is the code and datasets for the benchmark, rather than algorithm insights or results, the authors should release the code in a way that can be reviewed easily. It is easy to release anonymised repos with https://anonymous.4open.science/. The benchmark would also benefit from web-hosted docs.

**W3.** The paper's own novel baseline, HyMOPO, is noted to be unsuitable for the Walker2D and Hopper environments due to observation clipping issues, limiting its generality.

**Questions:**

**Q1.** Do you have an intuition as to why hybrid-RL algorithms scored worse than algorithms without the offline dataset? Do you think they could perform better if tuned properly, or do you think there is a fundamental problem?

**Q2.** Since baselines were drawn from different repositories, is it possible that the poor performance of some hybrid methods is due to suboptimal tuning for this specific hybrid task rather than a fundamental algorithmic flaw?

---

> ### Author Response · Authors · 2025-11-19
>
> We thank the reviewer for the thorough and constructive feedback. Your questions regarding the fundamental nature of the failure in hybrid RL (Q1 & Q2) inspired us to conduct a deeper analysis of the mechanism. We have added these findings to the revised version of the paper (section 4.3).
>
> To definitively answer whether this is a fundamental flaw or a tuning issue (Q1 & Q2), we investigated the mechanism of failure by analyzing the Value Estimation Error (predicted Q value / total return) during the rebuttal. We hypothesized that hybrid algorithms might over-trust the biased simulator.
>
> As shown in the table below, 'healthy' agents maintain a calibrated ratio of ~0.08. In the failure case our confounding challenge (challenge 4), this ratio spikes to 0.27—a >300% overestimation. This implies that the failure is fundamental: the agent suffers from signal dominance, prioritizing the 'clean' signal of the biased simulator over the 'noisy' signal of the true data. Tuning cannot fix this, as the agent is optimizing a fundamentally 'delusional' objective.
>
> | Algorithm | Challenge Type | Real Return ($G$) | Predicted $Q$ | Ratio ($Q/G$) |
> | :--- | :--- | :---: | :---: | :---: |
> | **HyMOPO** | Modeling Error (Challenge 1) | 11,500 | 882 | **0.077** |
> | **HyMOPO** | Low Confounding ($\sigma=0.01$) | 9,664 | 752 | **0.078** |
> | **SAC** | Hidden Dims (Challenge 2) | 2,254 | 178 | **0.079** |
> | **HyMOPO** | **High Confounding ($\sigma=0.05$)** | **1,538** | **395** | **0.269** |
>
>
>
> We will now address the weaknesses raised:
>
> **W1A:** We completely agree with the need for a standardized testbed. To address this, we have added Appendix B: The B4MRL Core Evaluation Protocol to the revised version of the paper. This protocol defines the exact matrix of environments, noise levels, and challenges required for a valid 'B4MRL-Verified' algorithm. This eliminates degrees of freedom and ensures future work cannot cherry-pick settings.
>
> **W1B:** We prioritized using authoritative community-standard implementations to ensure our baselines match the original authors' claims. However, to mitigate the 'confounding variables' of execution, our codebase includes a unified launching wrapper. This single entry point standardizes the evaluation loop, seed management, and data loading across all algorithms, ensuring that differences in performance are due to the algorithms themselves, not the evaluation harness.
>
> **W2:** Thank you for this point, while we attached the code in the supplementary material, it is indeed easier to read code in an anonymized repo. We updated the text to include the link (which is also available here:
>
> https://anonymous.4open.science/r/B4MRL-D8DD)
>
> **W3:** HyMOPO is intended as a representative baseline (adapting MOPO to the hybrid setting), not the paper's primary contribution. In addition, it’s not that HyMOPO cannot be run technically on Walker/Hopper, but that it fails completely on them. This is a known consequence of the observation clipping in these environments, which acts as a severe, non-ignorable confounding factor. The fact that HyMOPO fails here is in itself a scientific result: it demonstrates that without taking into account such discrepancies, algorithms lacking robustness mechanisms are prone to failure.

---

### Official Review · Reviewer_mnbH · 2025-10-31

**Soundness:** 3
**Presentation:** 3
**Contribution:** 3
**Rating:** 6
**Confidence:** 3

**Summary:**

This paper presents B4MRL (Benchmarks for Mechanistic Offline Reinforcement Learning), a comprehensive benchmark suite designed to evaluate offline-to-online RL algorithms when combining offline datasets with imperfect simulators in hybrid RL settings. The paper identifies four principal challenges in hybrid RL: (1) simulator modeling error (sim2real gap), (2) partial observability and state discrepancy, (3) action discrepancy, and (4) hidden confounding (offline2real bias). Unlike existing benchmarks (D4RL, VD4RL, ODRL, CARL, etc.), B4MRL uniquely addresses all four challenges simultaneously, providing a systematic framework for evaluation. Through extensive empirical evaluation on MuJoCo and Highway environments using online (TD3, SAC), offline (TD3-BC, IQL, MOPO), and hybrid (H2O, PAR-BC, HyMOPO) algorithms, the paper demonstrates a critical finding: current hybrid RL methods frequently fail to leverage both data sources synergistically, often performing worse than using either source alone, particularly when hidden confounding is present.

**Strengths:**

**Comprehensive and well-motivated problem characterization.** The paper articulates a compelling and timely problem: while offline RL and simulator-based RL are well-studied, hybrid methods that combine both remain poorly understood and underspecified. The four challenges (modeling error, partial observability, action discrepancy, confounding) are systematically presented with concrete real-world examples (healthcare, autonomous driving, recommender systems), making the motivation clear and accessible. The distinction between partial observability and hidden confounding (Section 2.4) is particularly valuable and often overlooked.

**Rigorous and modular benchmark design.** B4MRL provides a principled, composable benchmark architecture where challenges can be independently controlled and combined. The design choices are justified through ablation studies (Figure 4 demonstrates how to select which variables to hide), and the implementation details (Section B, Appendix) are thorough and reproducible. The use of parametric modifications to MuJoCo environments (gravity, friction, action noise) provides clean, interpretable ways to introduce discrepancies.

**Diverse algorithm coverage.** Evaluating eight different RL algorithms (online, offline, and hybrid) across multiple environments and challenges provides broad empirical coverage.​

**Weaknesses:**

**Hidden confounding implementation is somewhat simplistic.** The confounding benchmarks (Section 3, Challenge 4) introduce confounding by either adding noise or removing variables from observations, with the assumption that the data-generating agent saw the missing variables. This is a stylized form of confounding that may not capture the full complexity of real-world confounding scenarios (e.g., time-dependent confounding, continuous latent confounders). More sophisticated confounding mechanisms could strengthen the benchmarks.

**Narrow scope of environments and tasks.** All experiments use MuJoCo continuous control and one highway driving environment. The scalability and applicability to image-based observations, discrete action spaces, or more complex domains remain unclear. D4RL benchmarks are well-studied but represent a narrow slice of RL problems.

**Questions:**

**Q1:** Why is three seeds used instead of five or more? Was this a computational constraint? Can results be re-run with additional seeds for higher statistical confidence?

**Q2:** How sensitive are confounding results to the specific variables chosen to hide? Figure 4 shows variable importance varies across algorithms. Is there a principled way to select impactful confounders?

**Q3:** Can you provide theoretical characterization of when hybrid methods have fundamental limitations? Is there a theoretical explanation for why some combinations of challenges make hybrid methods perform worse than individual methods?

---

> ### Author Response · Authors · 2025-11-19
>
> Thank you for the thorough and valuable feedback, and for supporting our paper. We will now address the questions and the weaknesses raised:
>
> **Q1:** This was indeed due to computational constraints (the experiments currently reported cover ~1000 runs, excluding calibration). Additionally, the vast majority of our results show standard deviations under 5% of the mean, making the performance signals statistically significant even with three seeds. Furthermore, the performance gaps we identify (e.g., the >50% drop in challenge 4) are large enough that they are not sensitive to seed selection.
>
> **Q2:** You are correct that the confounding results are sensitive to the variable chosen. As known in causal inference, determining a priori which unobserved variable will cause the most confounding is notoriously difficult. Therefore, in B4MRL we selected the variables that empirically caused the largest performance drops, providing the most rigorous stress-test for the algorithms. Our code makes it easy to iteratively hide variables (via simple config changes) to identify the worst-case confounder (we made it easy also to hide multiple variables).
>
> **Q3:** This phenomenon arises from the fundamental conflict between the two data sources. In our confounding challenge (challenge 4), the offline data contains confounding (state noise), while the simulator contains modeling error (physics mismatch). Hybrid algorithms generally attempt to integrate these sources to estimate values or dynamics. However, because the sources imply different transitions for the same action, the algorithm receives conflicting supervision.
>
> To characterize this limitation, we empirically investigated the mechanism of failure during the rebuttal. We analyzed the value estimation error (predicted Q / total return) of HyMOPO on our confounding challenge (challenge 4). We found that in high-confounding settings, agents overestimate returns by >300% (a ratio of ~0.27 vs the expected ~0.08). This suggests a fundamental signal dominance failure: the agent optimizes for the simulator's clean but biased physics and effectively ignores the noisy but grounded offline data. This confirms that without a mechanism to arbitrate conflicting supervision, hybrid methods are theoretically bounded by the bias of the simulator. We added these findings to the paper in (section 4.3).
>
> **W1:** While real-world confounding is indeed complex, our results show that current SOTA methods fail even on simplistic confounding. To validate this, we evaluated IQL with frame-stacking (history) on our confounding challenge (challenge 4) during the rebuttal. The agent recovered only 46% of expert performance (score: 43.8 vs 94.3). This implies that even mechanistic noise acts as a structural barrier that modern recurrent architectures cannot resolve. If algorithms cannot pass these 'simple' unit tests, they are unlikely to succeed in complex, real-world scenarios.
>
> **W2:** We prioritized state-based continuous control to ensure diagnostic isolability. In high-dimensional visual domains, dynamics errors and representation errors are often entangled. B4MRL allows researchers to isolate the exact source of failure (e.g., distinguishing a 10% friction error from sensor noise), enabling precise debugging of algorithmic failure modes independent of representation learning.

---

> > ### Comment · Reviewer_mnbH · 2025-11-21
> >
> > Thank you for taking the time to put together a thorough response.
> > Given my overall assessment of the paper, my score will remain unchanged at this time.

---

### Official Review · Reviewer_4CTZ · 2025-11-02

**Soundness:** 2
**Presentation:** 2
**Contribution:** 1
**Rating:** 2
**Confidence:** 4

**Summary:**

The authors introduce a new benchmark for offline RL with biased offline data and imperfect simulators. Unlike standard benchmarks in offline RL (D4RL, etc.), they focus on realistic challenges such as modeling errors, causal confusion, and partial observability. Across four high-level challenges, they propose diverse variants of MuJoCo tasks (halfcheetah, hopper, and walker2d) and the Highway task. They benchmark several representative offline and online RL algorithms on this benchmark, showing that there is still room for improvement in addressing these challenges.

**Strengths:**

* The problem posed in this paper is sensible. I believe the community can benefit from offline RL benchmarks that incorporate more realistic challenges encountered in the real world.
* The paper is well-organized and easy to understand.

**Weaknesses:**

* The main weakness of this benchmark is its relevance. The authors motivate this benchmark from diverse real-world challenges in offline RL, such as dynamics discrepancy, causal confusion, and partial observability. However, the quality of the environments provided in this benchmark is limited in realizing these challenges. Specifically, the environments are limited to simple 2-D MuJoCo tasks (halfcheetah, hopper, and walker2d) and the (highly simplified) Highway environment. I'm unsure how impactful and useful these tasks are for today's offline RL research. From the motivation in the Introduction, I expected much more realistic benchmarks, such as datasets collected from actual human demonstrators or at least more realistic scripted or non-Markovian policies on more relevant tasks (e.g., complex and realistic robotic manipulation, long-horizon navigation, computer games, etc.).
* Moreover, while the individual challenges listed in Table 1 are sensible, they are implemented in a highly contrived manner. For example, the authors simply change the gravity or friction parameter to simulate modeling errors, and add Gaussian noise to simulate state/action discrepancies or causal confusion. In the real world, these challenges are often much more subtle -- such errors or noises are typically temporally correlated, non-Markovian, and biased. I'm not sure how representative and realistic the challenges implemented in this benchmark are. I also believe these implementations are too simplistic to be impactful enough as a standalone benchmark. I'd be fine with such simplifications in methodology papers, but as a benchmark paper, I think the bar should be higher than that of typical experiments in such papers.

**Questions:**

* Why is the benchmark called "mechanistic" offline RL?

---

> ### Author Response · Authors · 2025-11-19
>
> We thank the reviewer for the candid feedback. We appreciate the reviewer's push for high-fidelity benchmarks. We respectfully argue, however, that complexity is not the only metric for utility. We designed B4MRL specifically as a diagnostic unit-test suite. The fact that SOTA methods fail on these 'simple' challenges (as we show in the experiments, and in the new results added during the rebuttal) demonstrates that it could be highly useful for the community to solve these fundamental issues in isolation before moving to complex, high-dimensional environments where failure causes are entangled and impossible to diagnose.
>
> **Regarding weaknesses:**
>
> The reviewer rightly points out that real-world errors are often more complex than the Gaussian noise or parameter shifts used in B4MRL. However, our empirical results indicate that the community is not yet ready for that complexity, because current SOTA Hybrid methods fail even on these "simple" challenges. During the rebuttal, we evaluated IQL with history on our confounding challenge (challenge 4). Even with history, the agent recovered only 46% of the expert performance (score: 43.8 vs 94.3). This implies that even 'simple' Gaussian confounding acts as a structural barrier that modern recurrent architectures cannot resolve. If algorithms fail here, they will almost certainly fail on the complex, temporally correlated noise datasets. We see B4MRL as a necessary gatekeeper, where algorithms must pass these fundamental "Unit Tests" before tackling complex human manipulation.
>
> Furthermore, the controlled simplicity of B4MRL allowed us to empirically isolate a fundamental failure mechanism during the rebuttal: value hallucination. By separating the noise (Challenge 4) from the physics error (Challenge 1), we were able to measure the agent's value estimation error (predicted Q value / total return). We observed that in high-confounding settings, HyMOPO overestimated its expected return by >300% (predicting high rewards while achieving low returns). This empirical evidence suggests that the algorithms fail because they prioritize the clean signal of the biased simulator over the noisy signal of the true data. This specific diagnosis would be difficult to isolate in a complex, unstructured environment. B4MRL thus serves as a necessary "Unit Test" to identify and solve these signal dominance issues before tackling complex human manipulation.
>
> We see B4MRL as valuable to practitioners specifically because of its controlled nature:
> 1. **Isolating variables:** In real-world data (e.g., human manipulation), modeling error, confounding, and partial observability are entangled. If an algorithm fails, we don't know why. In B4MRL, because we contrived the error (e.g., specifically changing Gravity), we know exactly why the agent fails.
> 2. **Algorithm debugging:** The simplicity allows researchers to "Unit Test" their algorithms. If an algorithm can't handle a 10% friction change, it shouldn't be deployed on a real robot.
> 3. **Standardization:** Currently, researchers often create ad-hoc perturbations to test robustness (e.g., custom noise implementations). This creates a reproducibility issue. By standardizing these perturbations into a fixed suite, B4MRL provides a common tool to measure progress on specific robustness metrics.
>
> We hope this clarifies the goal of the benchmark. While it does not simulate the full complexity of the real world, the fact that the baselines consistently fail these diagnostics suggests that B4MRL is sufficiently difficult to drive algorithmic progress. If algorithms cannot pass these 'simple' unit tests, they should not be deployed on complex real-world tasks. We believe that B4MRL provides the necessary standardized gatekeeper to help the community build the robust foundations necessary to eventually solve complex, real-world tasks.
>
> **Regarding Q1:** The term "mechanistic" refers to the hybrid nature of the setting: specifically, the integration of explicit mechanistic knowledge (encoded in the physics simulator) alongside empirical offline data.

---

> ### Comment · Reviewer_4CTZ · 2025-11-22
>
> Thanks for the response. While I appreciate the potential need for an "unit-test" benchmark, I'm still not convinced by the quality of the benchmark proposed in this work. The authors claim that some existing offline RL methods fail on the new benchmark. However, without further evidence, it is still unclear (1) whether these tasks are actually solvable in practice, (2) whether this failure stems from the inherent limitations of offline RL methods or simply from suboptimal hyperparameters/design choices, and *more importantly*, (3) whether these tasks are worth solving. In particular, I'm not convinced that research papers producing methods that perform well on these "toy" tasks (with Gaussian noise, etc.) would help address real offline RL problems, because the proposed tasks are too simplified and too detached from actual challenges.
>
> I generally agree with the authors that solving these toy tasks can be a good initial step before tackling harder ones. However, given that the tasks introduced in this work are mostly simple modifications of existing environments (e.g., D4RL), I believe these simple tweaks alone do not necessarily meet the bar of ICLR publication without a clear justification about why they are sufficiently relevant or useful for producing impactful research in offline RL. To address this point, having more realistic tasks and datasets seems essential to me, especially given the motivation presented throughout the paper. For these reasons, I'd like to maintain my original rating this time.

---

### Official Review · Reviewer_afxy · 2025-11-05

**Soundness:** 3
**Presentation:** 3
**Contribution:** 3
**Rating:** 4
**Confidence:** 4

**Summary:**

The paper introduces B4MRL, a new benchmark suite for evaluating reinforcement learning algorithms that combine offline data and simulators. The benchmarks address four real-world challenges: modeling error, state and action discrepancies, partial observability, and hidden confounding. Experiments show that current hybrid RL methods often fail, especially when offline data contains hidden confounders. The results highlight the need for more robust algorithms that can reliably integrate both data sources. The paper proposes the benchmark to test these algorithms.

**Strengths:**

- Originality: Proposes (to my knowledge) the first RL benchmark that systematically combines all four sources of sim2real and offline2real error.

- Quality: Has educational value in the taxonomy of challenges. Implements them in simulated benchmark settings.
- Quality: Provides careful experimental evaluation with many baselines across offline, online, and hybrid RL methods

- Clarity: Clearly motivated and systematic presentation

- Significance: Addresses an important issue in the RL community, that is important for real-world applications.

**Weaknesses:**

- W1: partial observability calls for methods for POMDPs. The methods used, such as TC3-BC, SAC etc are all using TD learning and rely on the Markov property. Methods that use Monte-Carlo returns (such as PPO) or methods that have recurrent architectures (policy and value functions) would be the natural algorithms to be tested.

- W2: The hidden confounder problem is present in modern sim2real pipelines with teach to student distillation and privileged information used for the teacher. The students are always recurrent networks, to perform latent state estimation. I think you benchmark is great, but I also believe the problems have been solved already in practice.

- W3: Limited diversity of tasks/environments: While MuJoCo and Highway are used, the benchmarks are centered on classic continuous control tasks and may not generalize to more complex domains such as vision-based control.

**Questions:**

- Q1: I am mostly concerned about the non-Markovianity. I suspect that the results will change quite drastically, if methods are used that are designed for partial observability. Would be very interesting to understand if the confounding is really such a strong problem then. Now, confounding is somewhat convolved with partial observability. Can you provide evidence that the observed phenomena persist?

- Q2: what happens if you use recurrent architectures? (or as a first approximation provide a history of 4 observations?

- Q3: Fig 5: what do you mean with "moving from simple modeling error to high-impact hidden confounding"? Is the drop when setting 1+4 vs the setting 1?



Comments:
- line 204: "and P (r = 1|z = 1, a = a0) = 1/6" should that not be z=0?
- line 294/295: "...Gaussian noise into the action implemented by the agent to the simulator’s present state...". What means action into simulator's state?
- line 377: $o'_{\text{sim}}$ sim should prob. be a subscript.
- Fig 5b: I think the labels are confusing and redundant. The x-labels already contain the challenge. (Maybe use the descriptive names instead of numbers). Reduce the number of markers to the actually different runs (so one line should have only one marker). Also, a clearer description of sigma, h and g would be good.

---

> ### Author Response · Authors · 2025-11-19
>
> We thank the reviewer for the constructive feedback. The suggestion to evaluate history based methods provided a valuable opportunity to empirically distinguish the confounding challenge (challenge 4)  from standard partial observability (challenge 2).
>
> **Re W1, Q1, Q2: Does history (frame-stacking) solve the confounding challenge?**
>
> To address the concern that the confounding challenge might be reducible to a standard POMDP, we evaluated IQL (which is our strongest offline baseline) on HalfCheetah with medium-expert dataset using frame-stacking (with k=4 frames) for both the "hidden dimensions" and "confounding noise" settings (corresponding with $\sigma_\text{high}$ and $h_\text{high}$ in the paper).
>
> The results (which we also added to the revised version of the paper, page 9, figure 5) demonstrate a fundamental structural distinction:
>
> | Challenge Type | Clean Baseline (Oracle) | No History (1-stack) | With History (4-stack) | Result |
> | :--- | :---: | :---: | :---: | :--- |
> | **Partial Obs. (Hidden Dims)** | 94.3 | 64.0 | **89.9** | Recovered (~95%) |
> | **Confounding (Mechanistic Noise)** | 94.3 | 28.3 | **43.8** | Failed (<50%) |
>
> As can be seen above:
> 1. **Recoverable dynamics (hidden dims):** Adding history allowed the agent to recover near-optimal performance on the hidden dimensions challenge. This confirms that when missing information is inferable from past dynamics (e.g., velocity), the problem acts as a standard POMDP and can be solved via memory.
> 2. **rreducible uncertainty (confounding):** In contrast, adding history yielded only marginal gains on the noisy dataset challenge (score of 43.8 vs 94.3), failing to recover the performance on the non-confounded dataset. This implies that the confounding in B4MRL is distinct from a standard, solvable POMDP. Unlike missing physical variables (which could be temporally correlated and thus inferable), the mechanistic noise acts as an unstructured, independent confounder at each timestep. Because this noise is not a function of the history, the true state remains unidentifiable regardless of context length, creating a structural discrepancy that makes it hard to resolve, even with history.
> 3. **Mechanism of Failure (Value Hallucination):** To understand why this gap persists for hybrid methods specifically, we further analyzed the agent's value estimates during the rebuttal. We found that in this high-confounding setting, hybrid agents overestimate the return by >300% (predicted Q value / total return of  approximately 0.27 vs the desired 0.08). This indicates that the agent is "hallucinating" value by trusting the biased simulator's clean signal over the confounded data's noisy signal. Suggesting this is a signal dominance failure, not just a memory failure.
>
> Therefore, these experiments validate the taxonomy of B4MRL, where the benchmark acts as a diagnostic tool. Some confounding benchmarks can be alleviated (or even solved) using memory, but some don’t. This shed some light on “how much” confounding each challenge holds. For challenges where the hidden confounding is stronger (like the noisy dataset), the practitioner might need to refer to causal inference to address the confounding problems instead of just making architectural changes.
>
> **W2:**
>
> Regarding the use of teacher-student distillation in Sim2Real: If we understand correctly, this solution relies on the Online Sim2Real setting (where a privileged teacher/simulator exists). B4MRL specifically targets the Offline RL setting, where we assume access only to a static dataset and a biased simulator, with no access to a "perfect teacher" or the true environment during training. In this constrained setting, privileged distillation is not applicable. However, as we demonstrate in our experiments, having a hybrid-RL algorithm that can enjoy both worlds is a good direction going forward.
>
> **W3:**
>
> While we agree that vision-based tasks add realism, B4MRL prioritizes "mechanistic" tasks (modifying gravity, friction, noise) to ensure interpretability. This design choice allows researchers to isolate the exact source of failure (e.g., differentiating "modeling error" from "confounder"), which is often entangled in high-dimensional visual domains.
>
> **Regarding comments and Q3:**
> * Q3 (Fig 5): Yes indeed. The drop occurs when moving from Challenge 1 (Modeling Error only) to Challenge 1+4 (Modeling + Confounding). We have clarified the caption.
> * Line 204: You are correct, we fixed $z=1$ to $z=0$.
> * Line 377: Corrected.
> * Line 294: We modified the text.
> * Fig 5b: We have updated Figure 5b to remove redundant legends and reduced markers to reduce clutter. The x-axis now explicitly names the challenges (e.g., 'Modeling Error', 'Confounding') rather than using indices.

---

### Author Response · Authors · 2025-11-19

We thank the reviewers for their time, effort, and valuable feedback. We are pleased to report that we have conducted the requested experiments, including frame-stacking and hybrid RL degradation analysis. These new results have not only addressed specific concerns but have further deepened the paper's insights, validating B4MRL as a rigorous diagnostic tool.

We have uploaded a **revised PDF** containing the following updates:
1. **New Section 4.3 (Page 10)**: Detailed analysis of failure mechanisms, including the new "Frame-stacking" (figure 5, page 9, note that the previous figure 5 is now figure 6) and "Hybrid-RL degradation" (table 3) experiments.
2. **Appendix B**: The Core Evaluation Protocol to ensure standardization.
3. **Clarity**: We have decluttered Figure 5b (which is now Figure 6b in the revised paper), and added small modifications as suggested by reviewers.

### **Summary of new experimental findings:**
**1. Frame-stacking analysis:**

To determine if the confounding in B4MRL is merely a solvable POMDP, we evaluated IQL with frame stacking (aggregating 4 frames of observations together) on two variants of the confounding challenge (see Figure 5, Page 9):
* **Hidden dimensions**: History recovers ~95% of expert performance. This confirms that when confounding is due to missing physical variables (e.g., velocity), it can behave as a solvable POMDP.
* **Mechanistic noise**: History recovers only ~46% of performance.
* **Conclusion**: This finding implies that the mechanistic noise in B4MRL acts as a structural barrier that was not resolved, distinguishing this confounding error from standard partial observability.

**2. Mechanism diagnosis for Hybrid-RL degradation:**

To understand why hybrid algorithms fail on the confounding challenge (and whether it is a fundamental issue), we analyzed the Value Estimation Error (ratio of predicted Q value and the total return G) of HyMOPO (see table below which is also table 3 on page 10 in the revised version).

| Algorithm | Challenge Type | Real Return ($G$) | Predicted $Q$ | Ratio ($Q/G$) |
| :--- | :--- | :---: | :---: | :---: |
| **HyMOPO** | Modeling Error (Challenge 1) | 11,500 | 882 | **0.077** |
| **HyMOPO** | Low Confounding ($\sigma=0.01$) | 9,664 | 752 | **0.078** |
| **SAC** | Hidden Dims (Challenge 2) | 2,254 | 178 | **0.079** |
| **HyMOPO** | **High Confounding ($\sigma=0.05$)** | **1,538** | **395** | **0.269** |

**Result:** In "healthy" runs, the agent maintains a calibrated ratio of ~0.08 (which makes sense considering the discount factor). In high-confounding settings, the agent overestimates returns by a ratio of 0.269, over 3 times the calibrated ratio.

**Conclusion:** This isolates the failure mode as signal dominance: the agent optimizes the simulator's "clean" (but biased) physics while rejecting the offline data's "noisy" (but true) signal. This level of diagnostic precision validates the utility of B4MRL's controlled design.

**3. Standardization**

To prevent "cherry-picking" in future work, we have added Appendix B: The B4MRL Core Evaluation Protocol, which formally defines the exact matrix of environments and noise levels required for a valid benchmark run.

We believe these revisions directly address the reviewers' concerns regarding the benchmark's utility and rigor. We are happy to answer any further questions.

---

### Meta-Review · Area_Chair_Xgjj · 2026-01-06

**Summary:**

**Paper Summary:** This paper studies offline RL in hybrid setting using both an imperfect simulator and a noisy offline RL dataset. This is an increasingly more studied setting since it is difficult to get both large quantity of perfect offline RL dataset, or design a perfect simulator, and so a hybrid approach that can combine their benefit has been studied. The main interesting find in this paper is that under certain type of noise hybrid RL approaches underperform approaches that use only one of the artifacts.

Additionally, the paper enumerates challenges in offline RL. Experiments are performed on mujoco tasks with errors introduced by varying parameters. The resultant benchmark is called B4MRL and has both a noisy simulator and offline RL dataset.

**Reviewer Summary:** The main reviewer concern that informs my decision is lack of realistic experiments. Authors have made a claim in defense that B4MRL can act as a gatekeeper in identifying flaws in methods. While this is accurate, it is entirely possible that methods can succeed on realistic tasks while failing on basic tasks. In fact, this is true for LLMs already where they are quite successful at common human tasks while failing on more basic symbolic logic tasks (see "A Surprising Failure? Multimodal LLMs and the NLVR Challenge
", Wu et al., 2024). It is conceivable that real-world tasks might be hard in some sense such as observational complexity, while being simpler in other sense.

For this reason, I am leaning towards rejection despite useful claims in this paper, and the importance of the question that this paper seeks to answer.

**Reviewer Concerns:**

Main reviewer concerns raised were:

1. Experiments are not realistic, challenging, or up-to-date. In particular, reviewers mentioned lack of image-based evaluations (reviewer mnbH), lack of more baselines (reviewer afxy), and contrived nature of noise (reviewer 4CTZ). Authors have responded with experiments on

- frame stacking showing that partial observability alone is not the challenge
- experiments to explain why hybrid RL methods fail showing failure in accurate value estimation

These experiments are useful specially the value estimation result is quite insightful. However, authors did not include more domains or realistic experiments. So, the main concern here is not resolved.

2. Reviewers asked why hybrid RL approach was not working. Reviewers found that hybrid RL approaches incorrectly estimate the V-value by over-relying on the clean simulator. This concern should be resolved.

3. One reviewer asked for the repo of code which authors did. This concern is resolved.

**Reviewer Scores:**

This is a high-variance estimate.

1. Reviewer afxy: Their main concern are about non-Markovnity and recurrent experiment which the authors have addressed with new experiments. I think this reviewer would have increased their score from 4 to 6.

2. Reviewer 4CTZ: Their main concern are with relevant and contrived nature of noise which remain unaddressed. I believe they would keep their score at 2.

3. Reviewer mnbH: Their main concern on stylistic nature of noise and narrow scope of task remain unaddressed. I believe they would keep their score at 6.

4. Reviewer 5Jp9: Their main concern were standardizing a few settings. Authors have provided an anonymous repo and some standard setting. They have a more meta concern that there is not enough here for an ICLR paper. This concern is also shared by reviewer 4CTZ . Overall, they would have raised their score from 2 to 4.

This leads to final score of 2, 4, 6, 6.

---

### Decision · Program_Chairs · 2026-01-26

Reject